# Guidance on how to improve vertical covariance localization based on a 1000-member ensemble

Tobias Necker[1], David Hinger[1], Philipp Johannes Griewank[1], Takemasa Miyoshi[2], and Martin Weissmann[1]

[1]Institut für Meteorologie und Geophysik, Universität Wien, Vienna, Austria
[2]RIKEN Center for Computational Science, Kobe, Japan

**Correspondence:** Tobias Necker, tobias.necker@univie.ac.at

**Abstract.** The success of ensemble data assimilation systems substantially depends on localization, which is required to mitigate sampling errors caused by modeling background error covariances with undersized ensembles. However, finding an optimal localization is highly challenging as covariances, sampling errors, and appropriate localization depend on various factors. Our study investigates vertical localization based on a unique convection-permitting 1000-member ensemble simulation. 1000-member ensemble correlations serve as truth for examining vertical correlations and their sampling error. We discuss requirements for vertical localization by deriving an empirical optimal localization (EOL) that minimizes the sampling error in 40-member sub-sample correlations with respect to the 1000-member reference. Our analysis covers temperature, specific humidity, and wind correlations on various pressure levels. Results suggest that vertical localization should depend on several aspects, such as the respective variable, vertical level, or correlation type (self- or cross-correlations). Comparing the empirical optimal localization with common distance-dependent localization approaches highlights that finding suitable localization functions bears substantial room for improvement. Furthermore, we examine approaches for achieving positive semi-definiteness for covariance localization that hardly affect the sampling error reduction. Finally, we discuss the gain of combining different localization approaches with an adaptive statistical sampling error correction.

## 1 Introduction

The accuracy of the initial conditions provided by data assimilation systems strongly determines the skill of numerical weather prediction (NWP). Data assimilation (DA) relies on accurate estimates of forecast errors and error covariances that determine the weighting and spreading of observational information. However, modeling suitable error covariances is intrinsically difficult given various atmospheric processes acting on different scales, leading to situation- and flow-dependent error covariance structures. A breakthrough in estimating background errors has been the development of ensemble and hybrid data assimilation algorithms (e.g., Evensen, 1994; Bonavita et al., 2016; Bannister, 2017).

Considering the large state space of atmospheric models with a hundred million or more degrees of freedom, estimating error covariances with an ensemble forecast is demanding. Computational restrictions usually limit the number of affordable ensemble members to about 20 to 80 members (Bannister, 2017; Gustafsson et al., 2018). Ensemble systems, therefore, suffer

from severe under-sampling and sampling errors. For this reason, all ensemble and hybrid data assimilation systems require some form of sampling error correction for horizontal and vertical covariances, usually referred to as localization. Localization mitigates spurious correlations that arise from under-sampling. During the assimilation procedure, spurious correlations lead to sub-optimal analysis increments, resulting in a sub-optimal analysis and forecast as well as an inaccurate representation of forecast error by the ensemble. Horizontal and vertical localization are both challenging topics. Since fundamentally different processes are acting in the horizontal and vertical direction, the two structures require different solutions. Depending on the specific data assimilation algorithm, localization may also be important for other reasons, such as computational efficiency or rank deficiency. However, in this study, we focus on mitigating sampling errors independent of algorithm-specific constraints.

In the past decade, advanced high-performance computing systems such as the Japanese K-computer (Miyoshi et al., 2015, 2016a, b) enabled the first atmospheric ensemble simulations with thousands of ensemble members that can provide reliable error covariances (Kunii, 2014; Miyoshi et al., 2014; Kondo and Miyoshi, 2016; Necker et al., 2020a). The assumption that such large ensembles provide covariances close to true covariances allows for investigating sampling errors in smaller subsets. Necker et al. (2020b), for example, evaluated a statistical sampling error correction method based on a 1000-member ensemble. Preceding studies used a similar approach, but with a smaller ensemble size or lower resolution (e.g., Hamill et al., 2001; Poterjoy et al.; Bannister et al., 2017). Wu et al. (2020), for example, showed the potential of a 256-member ensemble for studying sampling errors in a 40-member ensemble focusing on covariances of radar observations on convective scales. Our present study aims to guide advances in vertical localization by analyzing vertical error correlations and the empirical optimal vertical localization derived from the convection-permitting 1000-member ensemble simulation of Necker et al. (2020a, b).

In recent years, several approaches for vertical localization have been developed. The most frequently applied localization approach is a distance-dependent localization that dampens long-range correlations (e.g., Houtekamer and Mitchell, 1998, 2001; Hamill et al., 2001; Miyoshi and Yamane, 2007). For example, many data assimilation algorithms use the Gaspari-Cohn tapering function (Gaspari and Cohn, 1999), which has a cut-off at a defined distance to damp correlations depending on the spatial distance. However, long-distance vertical error correlations often have a physical meaning. Vertically, e.g., radiative effects of clouds, deep convection, or hydrostatic balance can cause relevant correlations. Inappropriate localization can therefore eliminate meaningful error correlations (Miyoshi et al., 2014; Kondo and Miyoshi, 2016) or cause imbalances in the initial conditions (Kepert, 2009; Greybush et al., 2011; Lei et al., 2015).

Several studies investigated different aspects of optimal localization but often focused on horizontal localization. These studies cover fundamental research on sampling errors and their correction (e.g., Anderson, 2007, 2012; Flowerdew, 2015). Besides, some studies discuss suitable tapering functions for localization (e.g., Gaspari and Cohn, 1999; Gaspari et al., 2006; Bolin and Wallin, 2016; Stanley et al., 2021). Distance-dependent localization always requires tuning of localization scales. Consequently, multiple studies aim to derive optimal localization scales and functions by minimizing the error in correlations or the subsequent analysis (e.g., Perianez et al., 2014; Anderson and Lei, 2013; Lei and Anderson, 2014; Kirchgessner et al., 2014; Flowerdew, 2015).

Localization approaches can roughly be grouped into two categories: Adaptive and non-adaptive approaches. Non-adaptive approaches apply fixed domain- or variable-uniform localization functions and scales that do not change with time. Adaptive

localization approaches, such statistical sampling error correction methods, enable a flow- or error correlation-dependent local-
ization (e.g., Anderson, 2007; Bishop and Hodyss, 2009a, b; Anderson, 2012; Ménétrier et al., 2015a, b). A promising adaptive
localization approach is the global group ensemble filter (GGF; Lei and Anderson, 2014). The GGF enables adaptive vertical
localization of satellite radiances (Lei et al., 2016, 2020). However, adaptive methods usually require additional computational
resources, which can be a limiting factor in operational applications.

Current regional NWP models exhibit a grid-spacing of a few kilometers, allowing an explicit representation of deep con-
vection (Bouttier et al., 2016; Hagelin et al., 2017; Gustafsson et al., 2018). Finding optimal localization scales or functions is
challenging, particularly for convection-permitting simulations (Michel et al., 2011; Ménétrier et al., 2014; Destouches et al.,
2021). In these simulations, correlations and sampling errors depend on strongly non-linear dynamics, the chaotic nature of
convection, and uncertainties in microphysical processes that all contribute to rapid error growth (Hohenegger and Schaer,
2007; Ménétrier et al., 2014; Wu et al., 2020). However, little knowledge exists on the structure of short-term forecast errors in
regions with atmospheric convection (Hu et al., 2022). Consequently, better understanding of optimal vertical localization for
convection-permitting simulations has the potential to improve forecasts of convective precipitation and related hazards.

This paper investigates how vertical error covariances should be localized based on an existing convection-permitting 1000-
member ensemble simulation (Necker et al., 2020a). Our study focuses on correlations instead of covariances as correlation
sampling errors are the main contributor to covariance sampling error (Anderson, 2012). We will investigate domain-uniform
vertical localization but will also partly address the potential of adaptive localization approaches by applying a statistical sam-
pling error correction (SEC Anderson, 2012, 2016). Furthermore, we will analyze vertical correlations and empirically derive
an optimal vertical localization that minimizes the sampling error in subsamples of the 1000-member ensemble. Since the
optimal localization matrix is not necessarily symmetric positive semi-definite (SPSD), we explore methods to ensure SPSD.
Our setup allows for general conclusions independent of a specific DA algorithm. Among different aspects of localization, we
will address the following research questions:

- How do vertical error correlations for humidity, temperature, or wind behave on average?

- How should we localize vertical error correlations from small ensembles?

- How much error reduction can be achieved with a domain-uniform vertical localization or by combining different localization
approaches?

The remainder of the paper is outlined as follows: Sect. 2 introduces the 1000-member ensemble, the experimental setup, and
the weather period. Furthermore, we explain how vertical correlations and the empirical optimal localization are derived from
the 1000-member ensemble using sub-sampling. Sect. 3.1 evaluates vertical correlations and the empirical optimal localization
for single variable pairs to explore requirements for a variable-dependent localization. In Sect. 3.2, we group variables and
correlations based on similar behavior to derive an empirical optimal localization for self-/ and cross-correlations. Sect. 3.3
evaluates the error reduction achieved by different localization approaches and settings. Sect. 3.4 discusses different methods to
ensure that the localization matrix is SPSD. Finally, we summarize our results in Sect. 4 and discuss implications for improving
vertical localization.

## 2 Methods and experiments

### 2.1 1000-member ensemble simulation

Our study uses an existing convective-scale 1000-member ensemble simulation described in detail by Necker et al. (2020a). The 1000-member ensemble applies the full-physics non-hydrostatic Scalable Computing for Advanced Library and Environment regional model (SCALE-RM) and the SCALE Localized Ensemble Transform Kalman Filter (SCALE-LETKF) DA system (Lien et al., 2017). Using an offline nesting approach, the 1000-member ensemble setup couples two domains with different horizontal resolutions. Ensemble forecasts in the outer domain covering Central Europe (15-km grid spacing) delivered the boundary and initial conditions for the convective-scale ensemble forecasts in the inner domain covering Germany (3-km grid spacing). High-resolution short-term forecasts from the inner domain will be analyzed to evaluate correlations and localization.

Initial and boundary conditions: The data assimilation cycling has been performed in the coarse European domain assimilating conventional observations with a LETKF (Hunt et al., 2007). A set of 1000 independent and specifically constructed ensemble boundary conditions (BC) drive the European scale forecasts. These BCs combine 1000 climatologically scaled random perturbations with a 20-member analysis ensemble of the NCEP Global Ensemble Forecast System (GEFS). The GEFS 20-member analysis ensemble is used 50 times to reach 1000 BCs and afterwards combined with 1000 random climatologically scaled perturbations. This approach yields 1000 independent BCs that ensure sufficient ensemble spread when combined with relaxation to prior spread (RTPS Whitaker and Hamill, 2012). The boundary and initial conditions for the inner and convective-scale forecast domain are downscaled from 15 to 3 km resolution based on simulations in the European domain.

Our study uses the model output from the inner model domain with a $250 \times 230$ grid area centered over Germany with a 3 km horizontal resolution. This sub-domain excludes the Alps and regions within ten grid points to the domain boundary. The model output has 30 vertical levels ranging from the surface to the model top at 16.9 km. The original vertical grid is terrain-following and has fixed height levels above the surface (in m). Due to practical reasons, we extracted temperature (T), specific humidity (Q), and horizontal zonal (U) and meridional (V) wind components on 20 vertical pressure levels (100, 150, 200, 250, 300, 350, 400, 450, 500, 550, 600, 650, 700, 750, 800, 850, 900, 925, 950, 975 hPa). Performing our analysis on this modified grid allows horizontal averaging of data on pressure levels where needed. Overall, our examination includes ten short-term forecasts that have been initialized twice per day from 29 May to 2 June at 00 and 12 UTC, 2016. The 3-h forecasts valid at 3 and 15 UTC serve as a basis to compute and investigate background error correlations.

### 2.2 Weather period

Atmospheric blocking over the Atlantic influenced the large-scale flow over Europe in the five-day experimental period. The blocking led to a quasi-stationary weather pattern over central Europe with an upper-level trough over western Europe and a shallow surface low over central Europe. The low-pressure system was associated with a cold and a warm front that moved over Germany during the period. A convergence zone over southern Germany caused large-scale lifting. Furthermore, mid-level winds advected warm and moist air masses from southern Europe towards Germany at the beginning of the experimental period. Combined with the convergence zone, atmospheric conditions led to intense convection and heavy precipitation, including hail.

Weak pressure gradients and slowly moving convective cells resulted in high local precipitation rates and flash flooding. Due to these severe weather events, several studies focused on this exceptional period (e.g., Piper et al., 2016). Necker et al. (2020a, b), Nomokonova et al. (2022), and Craig et al. (2022) provide further details on the weather situation in this period as these studies also explore the 1000-member ensemble simulation with a different purpose.

## 2.3 Vertical localization

Error covariances are a key component in data assimilation and determine how assimilated information is weighted and distributed in state space. Given a sample of state vectors $x^n$ provided by a background forecast ensemble the flow-dependent sample error covariance matrix $\mathbf{P}$ can be computed as follows

$$\mathbf{P} = \frac{1}{N-1} \sum_{n=1}^{N} (x^n - \overline{x})(x^n - \overline{x})^T, \tag{1}$$

where $N$ is the ensemble size and $\overline{x}$ is the ensemble mean state. The covariance matrix $\mathbf{P}$ per definition is a symmetric, positive semi-definite matrix with variances on its diagonal and covariances on its off-diagonal entries. Each off-diagonal element contains a sample covariance $cov$ of two state variables $x_i$

$$cov(x_1, x_2) = r(x_1, x_2)\sigma(x_1)\sigma(x_2) \tag{2}$$

where $r \in [-1, 1]$ is the sample correlation and $\sigma$ the sample standard deviation.

Usually, the number of affordable ensemble members is limited in NWP due to a huge state space and computational restrictions. This deficit causes severe sampling errors. Consequently, all ensemble filters require a correction of sampling errors, often referred to as localization. For example Anderson (2012) highlighted that the sampling error in covariances is dominated by sampling error in the sample correlation $r$, not by sampling error in the variances. Therefore, our analysis will solely tackle sampling errors in sample correlations. Sample correlations are normalized with standard deviations and possess no unit. The normalization allows comparing or combining correlations of different variables facilitating the interpretation.

The implementation of localization depends on various factors determined by the type of ensemble filter. Usually, localization is applied directly to the background error covariance matrix using a Schur-product

$$\mathbf{P}_{loc} = \mathbf{C} \circ \mathbf{P}, \tag{3}$$

where $\mathbf{C}$ is the localization matrix (Gaspari and Cohn, 1999; Bannister, 2008). The matrix $\mathbf{C}$ consists of tapering factors $\alpha$ that are determined using the localization approach of choice. Based on the Schur product theorem (Horn and Johnson, 2012, theorem 7.5.3), positive semi-definite matrices $\mathbf{C}$ and $\mathbf{P}$ guarantee positive semi-definiteness of the localized covariance matrix $\mathbf{P}_{loc}$. Furthermore, localization matrices should feature ones on the diagonal similar to a correlation matrix to avoid undesired inflation of variances (Flowerdew, 2015).

| Variable | | Temperature (T) | Humidity (Q) | Zon. Wind (U) | Mer. Wind (V) |
|---|---|---|---|---|---|
| Temperature | (T) | TT | TQ | TU | TV |
| Humidity | (Q) | QT | QQ | QU | QV |
| Zon. Wind | (U) | UT | UQ | UU | UV |
| Mer. Wind | (V) | VT | VQ | VU | VV |

**Table 1.** Analyzed correlation pairs. Self-correlations on diagonal and cross-correlations on off-diagonal of the table. The first variable of each pair represents the ensemble at the reference level.

### 2.3.1 Distance-dependent localization

The most common localization approach is a distance-dependent localization that determines tapering factors $\alpha$ based on distance (Houtekamer and Mitchell, 1998, 2001). The vertical separation distance in our study is defined in $ln(p)$. We consider the widely used Gaussian-shaped Gaspari-Cohn function (GC; Eq. 4.10, Gaspari and Cohn, 1999) for comparison with other methods. Applying a GC function always requires the selection of the separation distance but guarantees positive definiteness. The separation distance is often referred to as the localization scale, while the cut-off radius is usually twice the localization

scale. In our study, we apply vertical localization according to the definition of Deutscher Wetterdienst (DWD) (Schraff et al., 2016). For DWD, the localization scale is determined by a pre-selected localization length that is multiplied by a factor of $(\sqrt{10/3})$. Operationally, the localization length of DWD is height-dependent and increases linearly in $ln(p)$ from the surface (0.075) to 300 hPa (0.5).

In Sect. 3.3, we apply three different domain-uniform GC localization setups: a) "*GC*": An optimally tuned GC localization

scale that applies a uniform localization scale for all variables and heights. b) "*GCLEV*": An height-dependent optimally tuned GC localization scale that is uniform for all variables. c) "*DWD*": A localization setting similar to DWD as described above that is also domain- and variable-uniform.

### 2.3.2 Sampling error correction (SEC)

Necker et al. (2020b) showed that an adaptive statistical sampling error correction (SEC, Anderson, 2012, 2016) substantially

reduces the sampling error in sample correlations and ensemble sensitivities. The SEC is a look-up table-based approach and corrects the overestimation of correlations caused by sampling noise. The look-up table is computed offline and based on Monte Carlo simulations that consider the likelihood of the correlation $r$. The SEC depends on the sample correlation, the ensemble size, and the assumed prior distribution of correlations. Here, we assume a uniform default prior distribution and apply the SEC table that is provided within the Data Assimilation Research Testbed (DART; Anderson et al., 2009). In Sect. 3.3, we will

compare the benefit of the SEC with different localization approaches. The comparison includes combinations of the SEC with these approaches.

## 2.4 Sub-sampling and vertical correlations

The sampling noise expected for zero correlation estimates and sample size $N$ is $(\sqrt{N})^{-1}$ (Houtekamer and Mitchell, 1998). For the 1000-member ensemble (N=1000), this estimation yields a very small sampling noise of approximately $3\%$. In comparison, a 40-member ensemble reveals an expected sampling noise of approximately $16\%$. Throughout this study, correlations computed using the full 1000-member ensemble serve as truth ($r^{1000}$) for the interpretation of vertical correlations and the evaluation of sampling errors and localization in smaller subsamples of the full ensemble. We focus on vertical correlations and sampling errors in 40-member subsamples as this is a typical ensemble size applied by operational weather services such as, e.g., Deutscher Wetterdienst. Preceding studies applied a similar approach for studying sampling errors (Hamill et al., 2001; Poterjoy et al.; Bannister et al., 2017; Necker et al., 2020a, b).

The present study will adopt the sub-sampling approach from Necker et al. (2020a) and Craig et al. (2022). The 1000-member ensemble provides 25 randomly drawn 40-member subsamples without repetition of members (illustrated in Fig. 1 (a)). We assume that the 40-member sub-ensembles of the 1000-member ensemble statically have sampling errors similar to those independent 40-member ensemble EnKF systems would have. As mentioned above, we will analyze ten 3 h forecasts. This setup results in a sample of 250 ensemble forecasts with 40 members that we can compare to the ten ensemble forecasts with 1000 members. The model domain has $250 \times 230$ grid points yielding 57.500 vertical columns in our domain. We will, therefore, analyze approximately $11.5 \times 10^6$ true and $287.5 \times 10^6$ 40-member vertical correlation profiles per variable pair, accounting for all 20 reference levels. This data set allows robust statistical analysis of error correlations, but it should be noted that error correlations may differ for other periods and regions.

In the present study, we will analyze four prognostic variables: temperature (T), specific humidity (Q), zonal wind (U), and meridional wind (V). This setup yields 16 correlation pairs (Tab. 1) that we will inspect for different reference levels. Furthermore, we will group correlations in "self" (e.g., temperature-temperature as shown in Fig. 1) or "cross" (e.g., temperature-humidity) correlations for highlighting common behavior. Subsequently, we will use the correlation coding shown in Tab. 1. For example, "TQ" combines all temperature correlations from the reference level to specific humidity at all other vertical levels in a column. Throughout the manuscript, we will mainly present results for the U-wind component as conclusions for the V-wind component are similar.

**Example of vertical correlations** Fig. 1 (a) shows an example of vertical self-correlations of temperature (TT) from reference level 500 hPa to all other levels in a single random vertical column. The 1000-member correlation (also referred to as true correlation) is one at the reference level and drops to half after approximately 100 hPa vertical distance. Given the true correlation, the temperature at 500 hPa weakly correlates with the temperature in the boundary layer. This weak correlation is linked to cloud shadowing by mid-tropospheric clouds and resulting colder near-surface temperatures. Almost no correlation is visible to levels above the tropopause, which lies around 200 hPa. Most 40-member sample correlations strongly deviate from the true correlation, highlighting the severe under-sampling issue. Sampling errors appear to be larger with increasing distance and smaller correlation values. This behavior motivates most distance-based localization approaches with a predefined tapering

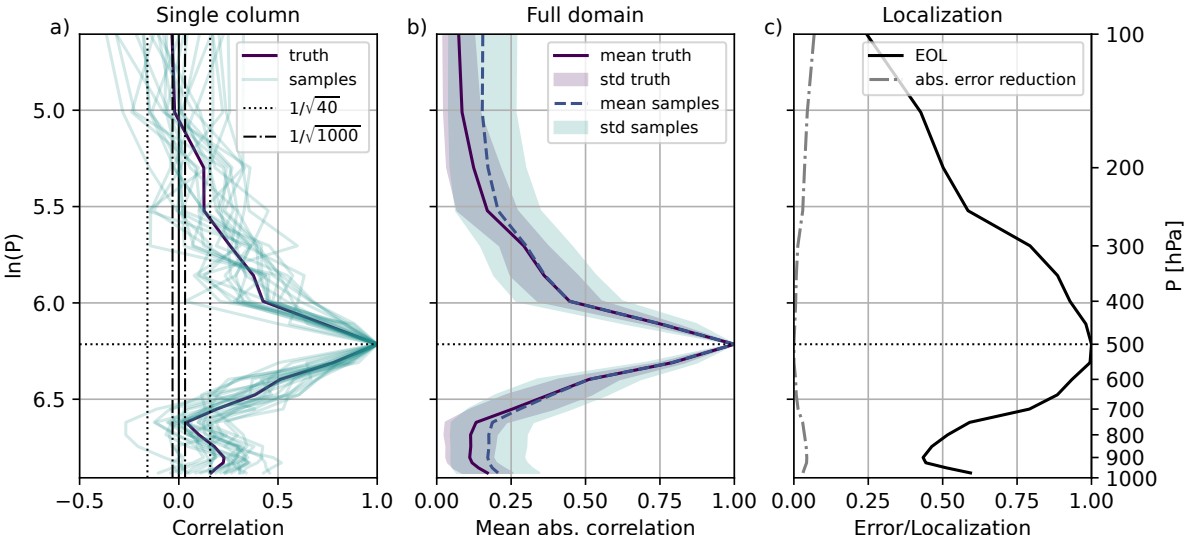

**Figure 1.** Vertical temperature-temperature correlations and empirical optimal localization for reference level 500 hPa on May 29, 2016, 15 UTC: (a) Single random column (b) Domain average. (c) Estimated domain-uniform EOL and absolute error reduction. The sample includes the correlations from all 25 40-member subsamples. Shading indicates spatial variability.

function that damp distant correlations. However, such an approach might cut off significant non-zero correlations, as seen for the boundary layer close to the surface in this example.

Throughout this manuscript we will analyze the 1000-member horizontally averaged absolute vertical correlation to support the discussion of the empirical optimal localization. Averaged absolute correlations are computed as follows:

$$\overline{r^{1000}}(t,z,p,A) = \frac{1}{K}\sum_{k=1}^{K}(|r_k^{1000}|), \tag{4}$$

where $K$ is the number of vertical columns in the domain. This analysis will be done separately for different forecasts $t$, reference levels $z$, pressure levels $p$, and variable pairs $A$.

Fig. 1 (b) displays an example of a mean absolute temperature self-correlations (TT) for reference level 500 hPa and a single date. On average, the mean absolute correlation of all 40-member subsamples well captures the shape of the true mean absolute correlation. However, 40-member ensembles overestimate the absolute correlation due to sampling error for weaker

correlations and larger distances. Furthermore, the 40-member correlations reveal a larger variance. Plotted in $ln(p)$, true and 40-member mean absolute correlations decay nearly symmetrically with increasing vertical distance from the reference level. This behavior explains why distance-dependent vertical localization scales are defined in logarithmic pressure coordinates.

## 2.5 Empirical optimal localization (EOL)

Our goal is to empirically find the optimal localization factor $\alpha$ that minimizes the sampling error or cost function $J$

$$J(\alpha, t, z, p, A) = \sqrt{\sum_{s=1}^{S}\sum_{k=1}^{K}(\alpha r_{s,k}^{40} - r_k^{1000})^2}, \tag{5}$$

where the minimization is done separately for each forecast time $t$, reference level $z$, pressure level $p$, and variable pair $A$. S is the number of 40-member ensembles ($S = 25$). This is equivalent to finding the $\alpha$ that minimizes

$$\sum_{s=1}^{S}\sum_{k=1}^{K}[\alpha^2 (r_{s,k}^{40})^2 - 2\alpha r_{s,k}^{40} r_k^{1000} + (r_k^{1000})^2]. \tag{6}$$

Taking a derivative with respect to $\alpha$ and finding the minimum gives us

$$\alpha = \frac{\sum_{s=1}^{S}\sum_{k=1}^{K} r_{s,k}^{40} r_k^{1000}}{\sum_{s=1}^{S}\sum_{k=1}^{K} (r_{s,k}^{40})^2}. \tag{7}$$

In other words, the empirical optimal localization (EOL) minimizes the Root Mean Square Difference (RMSD) between the 1000-member correlation and all 25 40-member sub-sample correlations for a chosen setting. For technical reasons, we minimized the cost function using the Brents method as implemented in scipy.optimize (Virtanen et al., 2020). Note that the range of localization is not confined to $[0, 1]$, which means that the EOL could increase correlations if required. Values larger than one can occur when the true correlation is larger than the sample correlation. For example, this can happen when estimating the EOL after applying other localization approaches. Negative EOL values can be observed when the EOL is computed for a small correlation sample (e.g., a single vertical column), which is not the case in the present study. However, we suggest setting negative EOL values to zero in case they might occur.

Applying the EOL by construction yields a symmetric but not necessarily a positive semi-definite localization matrix. In our case, constructed localization matrices were not positive semi-definite. Depending on the data assimilation algorithm, additional steps could be required to apply the EOL results to guarantee positive semi-definiteness of the localized covariance matrix. For this purpose, Sect. 3.4 will assess approaches for achieving positive semi-definiteness of the EOL. The assessment includes a numerical approach to approximate the EOL matrix with the nearest correlation matrix, which is SPSD.

Our approach for empirically estimating localization is inspired by Lei and Anderson (2014) who compare two methods: The Global Group Filter (GGF) and Empirical Localization Functions (ELF). The GGF minimizes the RMS difference between the estimated regression coefficients in subsets of the ensemble using a hierarchical ensemble filter (Anderson, 2007; Lei et al., 2016). ELFs are derived from an Observing System Simulation Experiment (OSSE) by minimizing the RMS difference between the true values of the state variables and the posterior ensemble mean (Anderson and Lei, 2013). In contrast to ELFs, the GGF and EOL purely judge localization based on ensemble sampling error without an OSSE. Furthermore, in contrast to the GGF, the EOL assumes the large ensemble correlation as truth for minimizing the sampling error. The EOL presented in our study corresponds to a non-adaptive distant-dependent domain-uniform vertical localization that is common for operational convective-scale regional data assimilation systems.

Figure 1 (c) displays the EOL $(\alpha(p))$ as estimated for the example of TT correlations introduced above and reference level 500 hPa. The domain-uniform EOL equals one at the reference level 500 hPa as no correction is needed. The EOL appears
broader and follows the shape of the mean absolute correlation. For example, this localization behavior was also described by Flowerdew (2015). The error reduction is largest for weak and distant correlations.

## 3  Results

This section presents mean absolute 1000-member vertical correlations and EOLs for various settings. First, we will evaluate how vertical localization for various single variable pairs should be constructed. Afterward, we will group variable pairs based
on similar behavior. Finally, at the end of the results section, we will evaluate the error reduction of all discussed localization approaches, including combinations with the SEC.

### 3.1  Vertical localization for single variable pairs

As discussed in Sect. 2.4, the domain-averaged absolute vertical correlation can aid the interpretation of the EOL. For this reason, we will first evaluate the mean absolute vertical correlation and then the EOL. Figure 2 shows the mean absolute
vertical correlation for all possible variable combinations and reference level 500 hPa. Self-correlations of the same variable all peak at the reference level. In contrast, cross-correlations are weaker and do not always exhibit a maximum correlation at 500 hPa. The TU correlation, for example, peaks around the tropopause, while the UT correlation reveals a minimum at that height. The mean vertical correlation length is variable dependent, shortest for specific humidity and longest for wind. The domain-averaged absolute vertical correlation only exhibits a fairly small variability within the five-day experimental period.
The variability between day to night also appears to be small (not shown). Results could, however, differ for other conditions or seasons, e.g., situations with strong atmospheric stability.

Next, we focus on the EOL derived for 40-member subsamples from all forecasts. Figure 3 displays the EOL for all variable combinations and reference level 500 hPa. The EOL depends on the prevailing correlation but has a different shape and vertical extent. As seen for the single-forecast example in Sect. 2.5, weaker correlations are more affected by sampling errors
and require stronger correction. Consequently, all cross-correlations require a stronger localization. The localization for cross-correlations reveals an amplitude smaller than one at the reference level. Given this behavior, tapering functions for cross-correlations should not be one at zero distance when applying a distance-dependent localization. Self-correlations are less affected by sampling error and require only a weaker correction, especially close to the reference level.

EOLs for humidity correlations all peak at the reference level 500 hPa (Fig. 3(a)). However, temperature and wind EOLs be-
have differently (Fig. 3(b,c)) and do not peak at the reference level following the correlation pattern. (Fig. 2(b,c)). For example, the TU EOL peaks around the tropopause, where winds are typically strongest. Wind correlations (e.g., UU; Fig. 3(c)) require only a small correction. The EOL for UV correlations is almost constant with height and does not show a distinct maximum. All self- and cross-correlations involving humidity peak at the reference level 500 hPa (for example, see UQ localization).

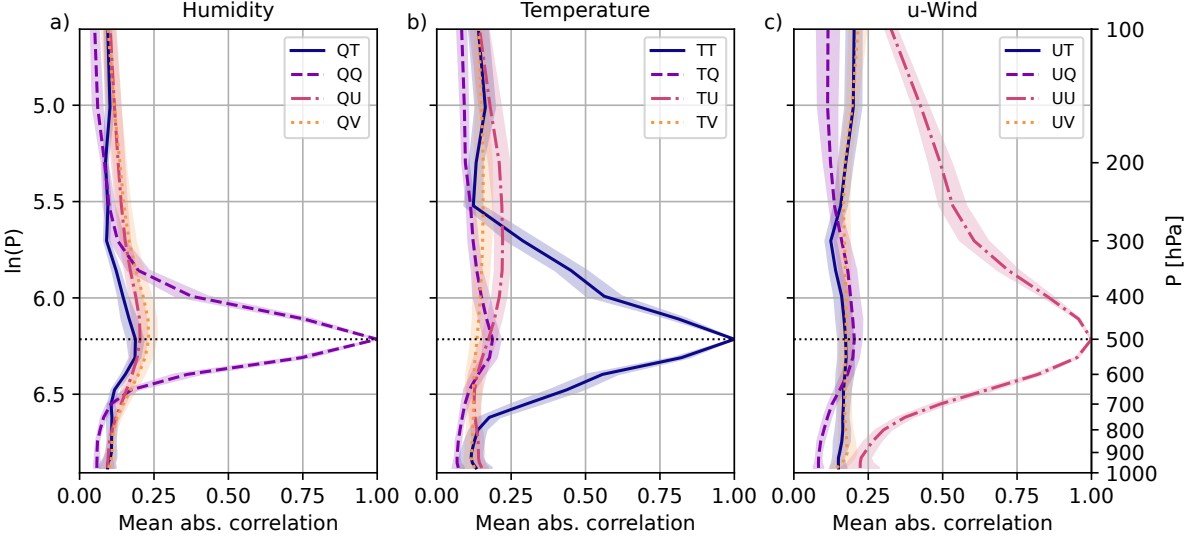

**Figure 2.** Domain averaged absolute 1000-member (true) vertical correlations for reference level 500 hPa and different variable pairs: (a) Humidity (b) Temperature (c) u-Wind. Mean and standard deviation over ten forecasts from May 29 to June 02, 2016.

Overall, the variability of domain-averaged correlations from forecast to forecast is small (Fig. 2). EOLs exhibit a larger variability than domain-averaged correlations. For most variables, the variability is larger close to the surface, especially for temperature correlations (Fig. 3(b)). Results should be treated with caution where changes of the EOL with height are smaller than the variability from forecast to forecast.

Subsequently, we will discuss the EOL for two additional reference levels to highlight changes in height within the troposphere. Figure 4 shows the EOL for a reference level 300 hPa. For reference level 300 hPa, EOLs appear to be broader compared to 500 hPa. This height-dependence is in line with larger vertical correlation length scales found for the upper troposphere in contrast to the lower troposphere, boundary layer, or close to the surface. Similar to other reference levels in the middle and upper troposphere, EOLs for correlations between wind and temperature reveal a maximum (TU; Fig. 4(b)) and minimum (UT; Fig. 4(c)) above the tropopause level.

All reference levels within the boundary layer show similar behavior of the EOL (see, for example, Fig. 5 using reference level 900 hPa). The EOL shows a narrow optimal vertical localization for reference levels close to the surface. In contrast to higher reference levels, also the EOL of cross-correlations peaks at the reference level (Fig. 5). The EOL drops to different constant values with increasing distance. For wind and humidity, the EOL reveals an almost constant value above 550 hPa. In contrast, the EOL for temperature steadily declines with increasing distance until the domain top. Temperature self-correlations (TT) exhibit a second peak in the upper troposphere. EOLs do not converge to zero for large vertical distances. Separation distances where the EOL converges to a small constant value could indicate suitable cut-off distances. A common aspect of the choice of cut-off distance is the signal-to-noise ratio that depends on the ensemble size and correlation strength.

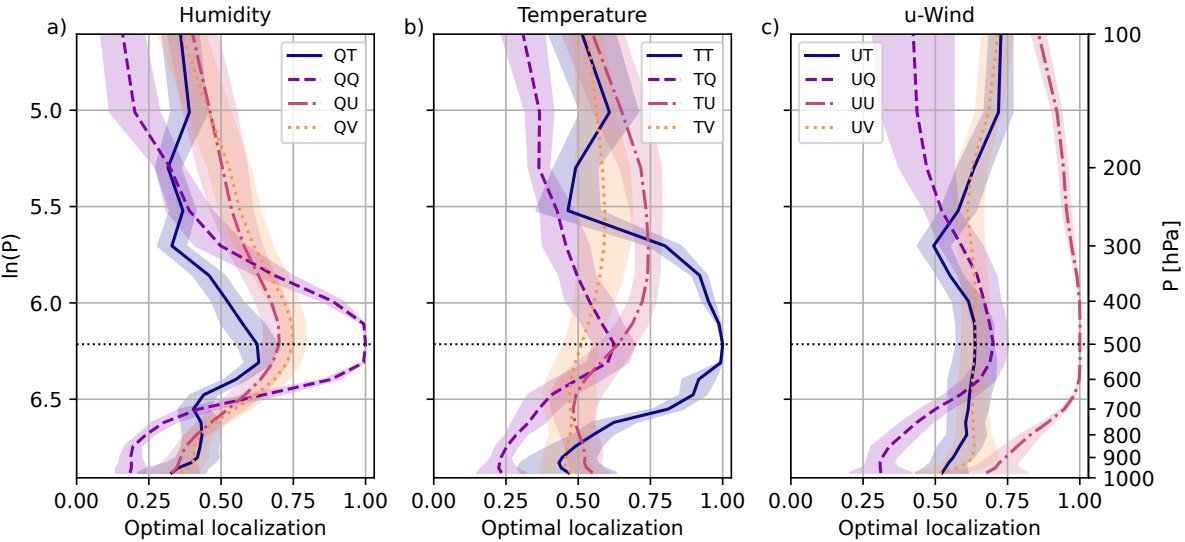

**Figure 3.** Empirical optimal localization (EOL) for vertical sample correlations of 40-member ensembles: (a) Humidity (b) Temperature (c) u-Wind. Mean and standard deviation over ten forecasts from May 29 to June 02, 2016.

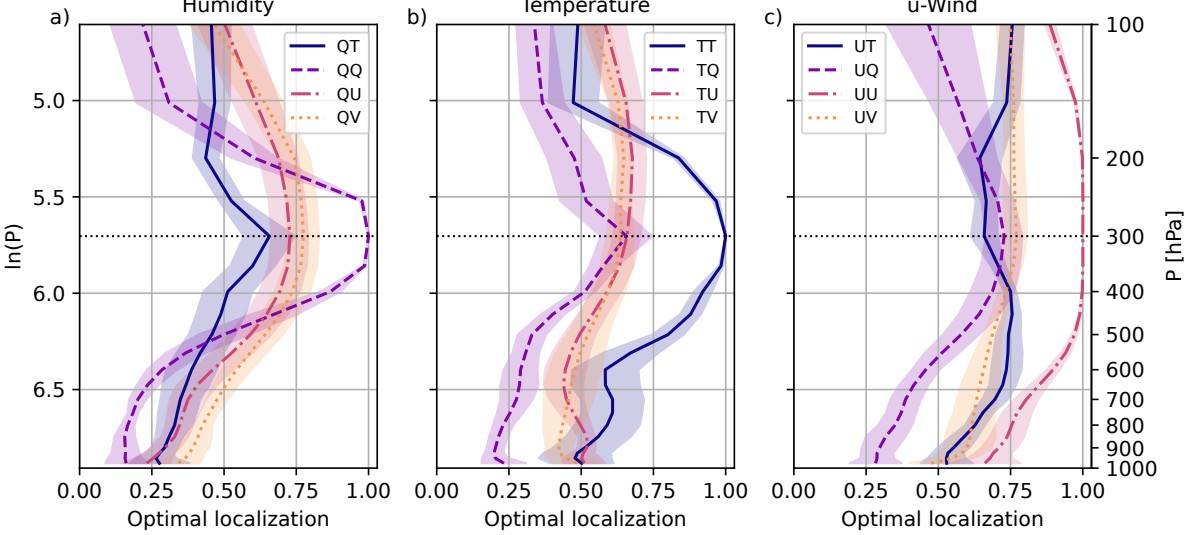

**Figure 4.** Same as Fig. 3 but for reference level 300 hPa.

### Error reduction for different variables

Assessing the EOL for single variable pairs revealed several requirements for vertical localization. Now, we evaluate the error reduction by the EOL, considering each possible correlation pair separately. The 1000-member ensemble correlation serves

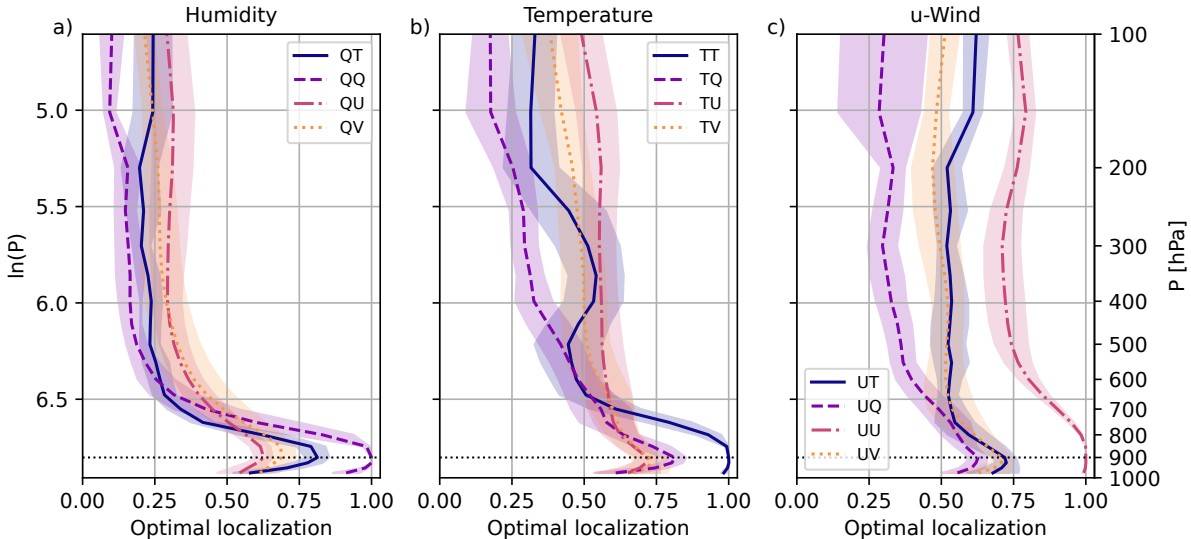

**Figure 5.** Same as Fig. 3 but for reference level 900 hPa.

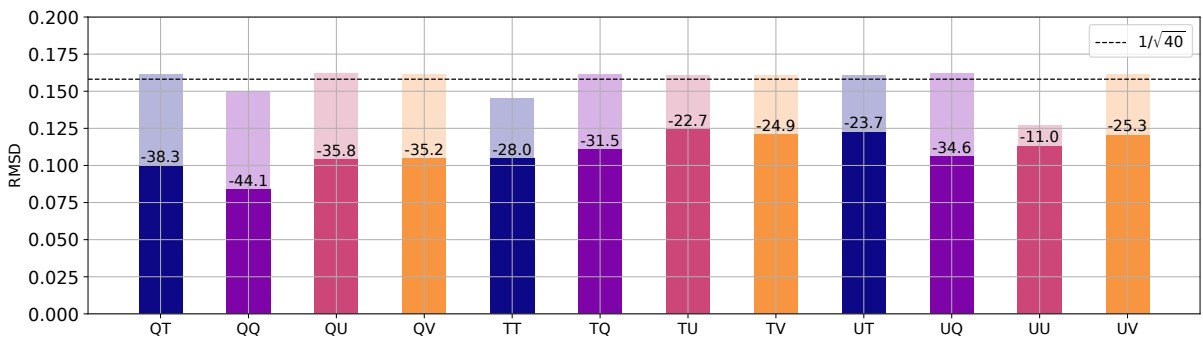

**Figure 6.** Root mean square difference before and after the EOL was applied to each vertical correlation. Shading and numbers (%) indicate the change in RMSD analysed for each variable pair averaged over all reference levels, columns, subsamples, and ten forecasts from May 29 to June 02, 2016. Self correlations are highlighted via hatching.

as truth to compute the RMSD of each 40-member sub-sample correlation. Figure 6 displays the RMSD before and after applying the EOL. The applied EOL varies for each forecast and height level for the error evaluation. The final result shows the average RMSD of all 40-member subsamples, forecasts, and height levels. The results can be interpreted as a benchmark of the maximum possible correlation error reduction achieved by a domain-uniform height and variable-dependent localization. Results for optimizing the analysis may lead to different optimal localization values under some circumstances, but this analysis

is beyond the scope of this paper.

The sampling error of the 40-member correlation of most correlations lies within the expected range and close to $(\sqrt{40})^{-1}$ (Fig. 6). Self-correlations exhibit a smaller sampling error as, on average, they are stronger and less affected by spurious correlations. The error reduction achieved by the EOL ranges approximately from 10 to 40%, depending on the variable pair. The QQ self-correlation benefits most from localization, whereas the UU self-correlation benefits the least. Correlations involving humidity are weaker and, therefore, benefit most from localization. On the other hand, correcting temperature correlations seems most challenging. Temperature correlations exhibit the largest RMSD, even after applying the EOL. The error is larger than for wind correlations, which is surprising considering a larger correlation strength and length for wind. This result could originate from a larger variability of vertical temperature correlations within the domain, given strong convective processes and associated latent heat release. Temperature correlations, consequently, could benefit from an adaptive localization that applies different localization scales within the domain depending on, e.g., vertical velocity. First tests showed promising results for such a situation-dependent approach, but a thorough evaluation will be left for subsequent study.

## 3.2 Vertical localization for grouped variable pairs

Some operational DA systems apply a uniform distance-based vertical localization that does not change with time, height, variable, or observation type. In this case, appropriate localization needs to meet several requirements using a suitable uniform localization approach. Results in Sect. 3.1 showed that cross-correlations systematically behave differently than self-correlations. For this reason, we will now evaluate the mean absolute correlation and EOL of three groups of variables: self, cross, or all correlations combined. Derived EOLs now minimize the sampling error for all gathered correlations of each group.

Fig. 7 displays the mean absolute correlation for the three groups of correlations. The results show the average correlation and its variability over the ten forecasts. Self-correlations again highlight the height dependence of the vertical correlation length and always exhibit a peak that is one. Cross-correlations are weaker and only exhibit a narrow peak at the reference level. For all correlations combined, the peak amplitude is closer to the peak of cross-correlations as there are more cross- than self-correlations. Combining all correlations or only cross-correlations results in a peak amplitude smaller than one at the reference level.

In contrast, the peak amplitude of the EOL for all correlations is closer to the peak of self-correlation (Fig. 8). The shape of EOLs substantially differs from the single variable pair cases. The EOL is weaker due to wind correlations that account for half of all correlations. The change in the shape of the EOL indicates that different tapering functions could be needed for different variables. Minimizing the error for grouped correlations, the strength of the EOL is always weaker than 0.4. Finally, domain averaged absolute correlations reveal a small variability from forecast to forecast (Fig. 7). The same applies to EOLs. Only the EOL of self-correlations exhibits a slightly larger variability, especially far from the reference level (Fig. 8).

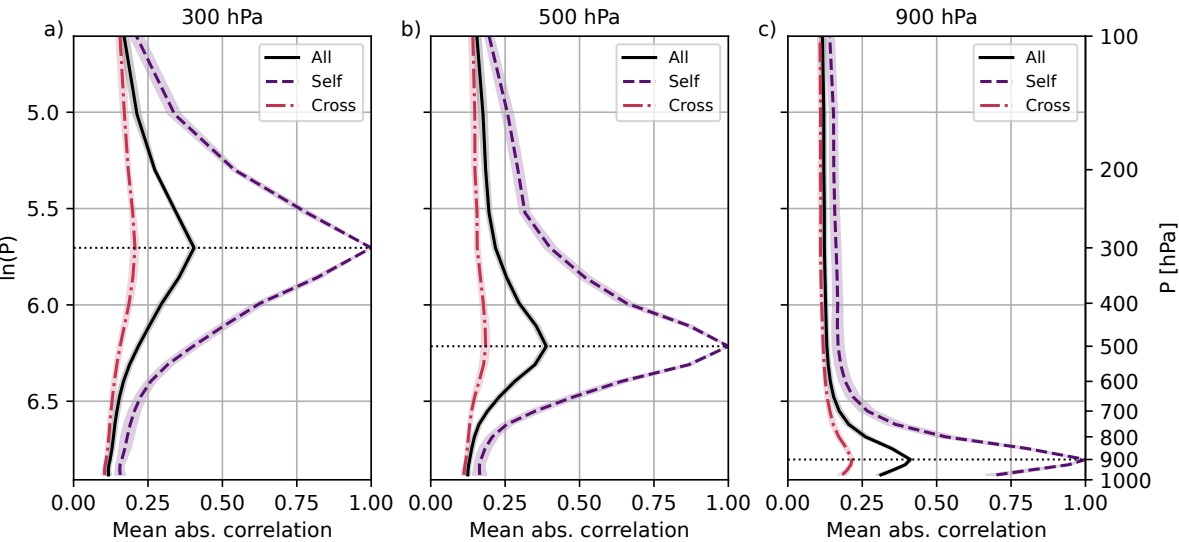

**Figure 7.** Domain mean absolute 1000-member (true) vertical correlations for different variable combinations (self, cross, and all): reference levels (a) 300 hPa (b) 500 hPa (c) 900 hPa. Mean and standard deviation over ten forecasts from May 29 to June 02, 2016. Note that the standard deviations are small and hardly visible.

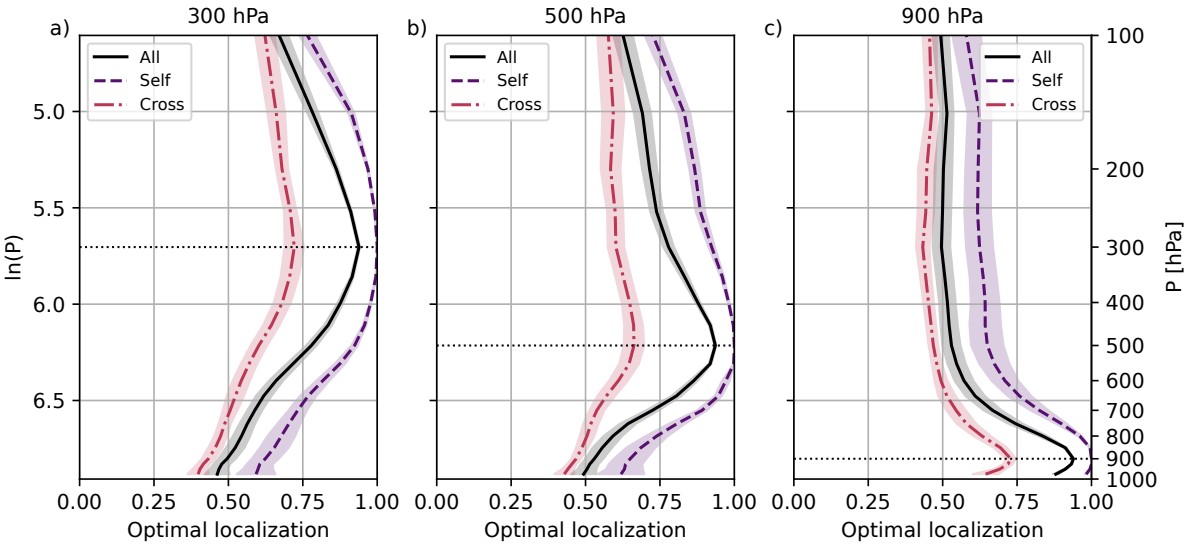

**Figure 8.** Empirical optimal localization (EOL) for vertical sample correlations of 40-member ensembles and different variable combinations (self, cross, and all): reference levels (a) 300 hPa (b) 500 hPa (c) 900 hPa. Mean and standard deviation over ten forecasts from May 29 to June 02, 2016.

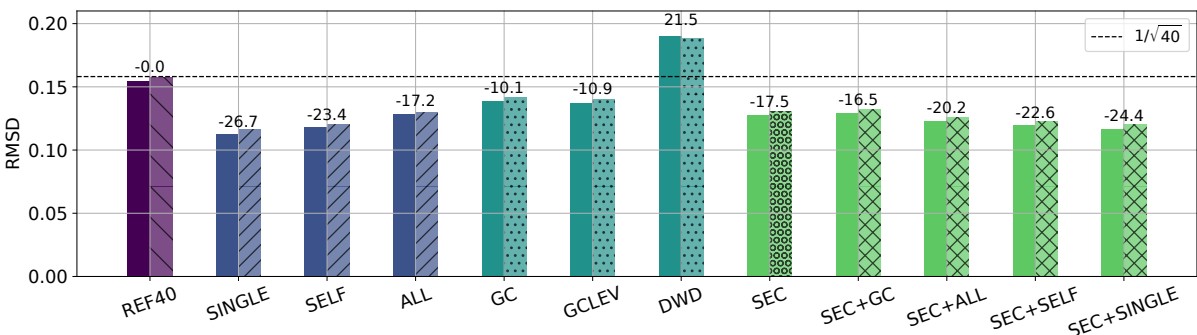

**Figure 9.** Root mean square difference before and after localization of 40-member vertical sub-sample correlations. EOL and Gaspari-Cohn scales are obtained and tuned using the first eight forecasts. Errors are evaluated using two independent forecasts on June 02, 2016: 3 UTC (opaque) and 15 UTC (hatched). Numbers (%) indicate the average change in RMSD analysed for different settings (x-axis labels).

## 3.3 Evaluation of error reduction

### 3.3.1 Setting

As discussed in Sect. 3.1, the maximum reduction of sampling errors achieved by an EOL ranges from 11 to 44 % depending on the variable pair. Now, we will compare the performance of the EOL with different localization setups that use two common localization approaches, a distance-dependent localization using a Gaspari-Cohn tapering function (GC; Houtekammer1998, Gaspari1999) and a statistical sampling error correction (SEC; Anderson 2012). Furthermore, we investigate the benefit of combining non-adaptive localization approaches with the adaptive SEC. Compared to Sect. 3.1, the improvement will be evaluated using 1000-member correlations from independent background forecasts. Again we will analyze the improvement relative to uncorrected 40-member ensemble sub-sample correlations (*REF40*, Fig. 9). The first eight forecasts (29th May to 1st June 2016) serve as training data for estimating EOLs. Similarly, localization scales for distance-dependent localization are tuned using the same training period. We then verified the performance using the last two independent forecasts on 2nd June 2016.

### 3.3.2 Empirical optimal localization (EOL)

Figure 9 displays the error reduction achieved by all considered vertical localization setups. *REF40* shows the RMSD found when modeling error correlations using small 40-member ensembles without localization. First, we will evaluate the performance of different EOL settings. Applying a different EOL for each variable pair and height (as presented in Sect. 3.1) gives the largest error reduction of all setups (*SINGLE*, 26.7 %). Only small differences are visible between day and night (not shown). Using different EOLs only for self- and cross-correlations leads to a slightly reduced performance but still gives about 23 % error reduction (*SELF*). Applying an EOL that was estimated for all correlations at once reduces the error by 17 % (*ALL*). Given these results, treating variable pairs, self- or cross-correlations differently enables substantial improvements. Finally,

we tested the error reduction for applying the EOL estimated for self-correlations to both self- and cross-correlations of each variable (e.g., EOL derived from TT applied to TT, TQ, TU, and TV). For this setting, the error reduction was similar to ALL or SEC (not shown), which underlines the benefit of treating self- and cross-correlations differently.

### 3.3.3 Distance-dependent localization

Now, we will compare the performance of EOLs to three different domain-uniform distance-dependent localization approaches using Gaspari-Cohn functions. Sec 2.3.1 provides more details on distance-dependent Gaspari-Cohn localization and details on all three considered localization setups (*GC*, *GCLEV*, and *DWD*). We will first evaluate two optimized setups with tuned localization scales (*GC* and *GCLEV*) and then compare them to a non-tuned setup (*DWD*).

*GC* uses a uniform localization scale for all levels and variable pairs, and *GCLEV* uses a height-dependent optimal localization scale that changes with the reference level. *GC* reduces the sampling error by about 10 %. Using height-dependent localization scales (*GCLEV*) slightly improves the performance further by about 1 %. However, the small gain of the height-dependent localization is partly associated with a sub-optimal shape of a Gaussian-shaped tapering function, given the error reduction achieved by the uniform EOL (*ALL*). This comparison highlights that finding suitable tapering scales and functions bears great potential for improving vertical localization.

In contrast, a vertical localization constructed similar to the regional DA system of Deutscher Wetterdienst (*DWD*) increases the difference of the 40-member ensemble correlation with respect to the 1000-member ensemble. The increased difference originates from the damping of meaningful error correlations. The DWD system employs a LETKF that uses observation-space localization, tuned to function in all seasons and weather situations that may differ from our investigation period. Furthermore, it needs to be considered that localization in the LETKF also affects the degrees of freedom of the analysis (Hotta and Ota, 2021). The DWD setup illustrates that an appropriate localization depends on various aspects. Consequently, our findings will likely have different implications for different DA algorithms. However, the LETKF, for example, could benefit from applying different localization scales for different observed variables. Based on the results in Sect. 3.1, humidity, temperature, or wind require different vertical localization scales.

### 3.3.4 Sampling error correction (SEC)

Now, we will evaluate the benefit of using a look-up table-based sampling error correction (SEC) that adjusts correlations based on predefined statistical assumptions. The SEC is an adaptive localization approach that corrects sampling errors as a function of the correlation value. Therefore, the SEC applies an individual correction for each correlation within the domain. An adaptive localization (*SEC*) achieves 17.5 % error reduction and outperforms a optimal domain-uniform GC localization. The SEC exhibits a similar error reduction as seen for *ALL* but can not outperform the *SELF* or *SINGLE* setup. An optimal domain-uniform localization can compete with an adaptive statistical sampling error correction for the evaluated period.

### 3.3.5 Combined approaches

Finally, we investigate the benefit of combining the statistical SEC with an EOL or a distance-dependent localization. For this analysis, EOLs have been estimated after applying the SEC to highlight the maximum error reduction achieved by combining SEC with an optimal localization. The localization scale of the distance-dependent localization is kept the same as for the *GC* setup to emphasize required changes for the localization scale. *SEC+GC* reveals a similar performance as the SEC alone but outperforms the *GC* setup. Combining *SEC* with *GC* requires a re-tuning of localization scales to larger values (not shown). Combining the SEC with a uniform EOL (*SEC+ALL*) reduces the sampling error by about 20 %. However, combining the SEC with the *SELF* or *SINGLE* EOL leads to less error reduction. The poor performance could originate from sub-optimal assumptions made in the derivation of the SEC (Anderson, 2016; Necker et al., 2020b). For example, the EOL exhibited values larger than one when estimated after applying the SEC, compensating for an over-correction of sampling errors, especially close to the reference level (not shown). In this study, we apply the most general SEC look-up table as provided in the Data Assimilation Research Test (DART; Anderson et al., 2009), which assumes that each correlation value is equally likely. Studying more informed prior assumptions in the SEC may lead to better results but is beyond the scope of the present study.

### 3.4 Covariance localization and positive semi-definiteness

The EOL approach empirically yields an optimal localization by minimizing differences between sample correlations and a defined true correlation. By design, the results discussed above exclude algorithm-specific requirements to better understand how vertical localization should behave in different situations. However, further steps might be required to apply EOL results depending on the data assimilation algorithm. As mentioned in Sect. 2.5, using EOL estimates does not necessarily yield a positive semi-definite localization matrix. Consequently, EOL localization could cause non-proper mathematical covariance matrices and undermine the data assimilation process. For example, positive semi-definite covariance matrices are elemental when solving a quadratic cost function as they ensure a global minimum. For this reason, we will now discuss different approaches that allow for achieving positive semi-definiteness of the EOL.

Ménétrier et al. (2015a) applied a Gaussian fitting based on the Gaspari-Cohn function (Gaspari and Cohn, 1999) to ensure SPSD. However, fitting a correlation function restricts the localization to a specific function shape. This restriction can diminish the error reduction. In our case, the error reduction achieved by an optimally tuned Gaspari-Cohn function is substantially smaller than by the EOL, as discussed in Sect. 3.3 and Fig. 9. This finding motivates exploring other localization functions (Daley et al., 2015; Bolin and Wallin, 2016) including multivariate localization functions (Stanley et al., 2021) to improve fitting approaches.

Besides function fitting, re-conditioning of matrices can help achieve positive definiteness (Tabeart et al., 2019). Given its aim, a localization matrix should have similar properties to a correlation matrix, which exhibits ones on the diagonal. Higham (2002, theorem 2.5) states that for a symmetric matrix with t non-positive eigenvalues and diagonal elements $\geq 1$, the nearest correlation matrix has at least t zero eigenvalues. Following this theorem, we attempted to set all negative eigenvalues to zero using an EOL-based localization matrix and eigendecomposition. Forcing negative eigenvalues to zero ensured positive semi-

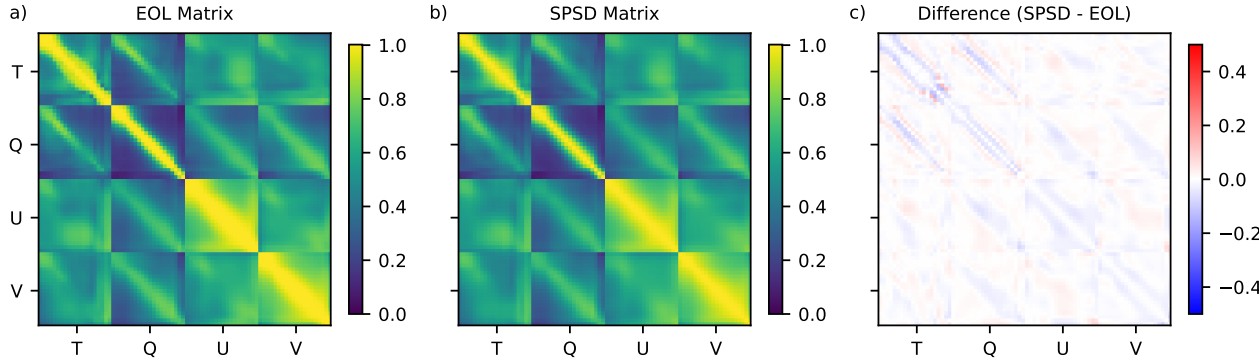

**Figure 10.** Examples of EOL-based localization matrices for a single vertical column: (a) matrix C constructed based on the *SINGLE* case, (b) resulting nearest correlation matrix C following the NCM algorithm, and (c) changes due to enforcing positive definiteness.

definiteness but led to values larger than one on the diagonal of the matrix. Further matrix transformations were required to achieve a proper correlation matrix.

Finally, our most successful attempt in achieving positive definiteness aiming for the least changes in the EOL was by searching for the nearest correlation matrix using specifically designed mathematical algorithms. For example, Higham (2002) highlighted that the convexity properties of the problem allows for finding a unique nearest correlation matrix (NCM) for a given symmetric matrix, in our case the EOL based localization matrix. The distance between matrices can be measured using weighted Frobenius norms. We apply the NCM algorithm (Higham, 2002) for the *SINGLE* EOL case as a proof of concept.

Employing four state variables on 20 vertical levels yields an $80 \times 80$ EOL-based vertical localization matrix for one vertical column (Fig. 10 (a)). The eigenvalues of this EOL matrix range from about -0.5 to 50, with approximately half the eigenvalues being negative. The EOL localization matrix features ones on the diagonal and reconfirms that cross-correlations require stronger tapering. The nearest correlation matrix computed by the NCM algorithm exhibits small changes on off-diagonal elements compared to the EOL-based matrix (Fig. 10 (b,c)). Zooming in, the SPSD matrix appears to be smoother. Comparing

the error reduction (not shown), the nearest SPSD matrix performs only marginally worse than the EOL matrix. The relative difference in sampling error reduction for a test case was lower than 1%. However, the EOL achieves the larger error reduction as it is designed to minimize the sampling error without any constraints.

This example suggests that ensuring SPSD can be achieved with minor changes in the EOL estimate. However, providing a general answer on how the EOL needs to be adapted is difficult as changes will depend on the construction of the localization

matrix and its unique nearest correlation matrix. NCM algorithms can iteratively determine the nearest correlation matrix for a symmetric matrix and could be a useful tool for data assimilation. Choosing the best approach to guaranteeing SPSD is likely case-dependent given changing properties of the problem and potentially very large matrices in NWP.

## 4    Conclusions and discussion

Current ensemble data assimilation systems suffer from severe under-sampling requiring vertical localization of error covariances. Our study analyzes vertical correlations from an existing convection-permitting 1000-member ensemble simulation (Necker et al., 2020a, b). The 1000-member ensemble correlation is assumed as truth for studying reliable vertical correlations and optimal vertical localization in 40-member subsamples. The unique convective-scale simulation covers ten forecasts in a five-day mid-latitude summer period. Our analysis includes four prognostic variables (humidity, temperature, and horizontal wind components) on 20 pressure levels. We apply the 1000-member ensemble and various 40-member subsamples to derive an empirical optimal localization (EOL) for different settings. Those settings include localization for single variable pairs and variables grouped by common behavior. Presented EOLs minimize the sampling error in sample correlations assuming the 1000-member correlation as truth, and provide insights on how to construct an optimal vertical localization independent of algorithm-specific constraints.

Furthermore, we use the 1000-member ensemble to evaluate the error reduction achieved by different localization approaches. These approaches include EOLs, distance-dependent localization approaches using a Gaspari-Cohn tapering function (Houtekamer and Mitchell, 1998; Gaspari and Cohn, 1999), and an adaptive statistical sampling error correction (Anderson, 2012). Overall, our results lead to the following conclusions for vertical localization:

– **Localization scales:** All investigated variables reveal different average correlation scales, which result in different EOL scales. Within the troposphere, EOL scales increase with height. Humidity requires the strongest localization with short scales. EOL scales for temperature appear to be larger than for humidity and exhibit the largest variability from forecast to forecast. Given a high variability, temperature correlations could benefit most from using adaptive localization. Our results indicate that winds can be vertically correlated throughout the troposphere, resulting in the largest localization scales. Given this outcome, it could be beneficial not to cut off wind correlations within the troposphere.

– **Localization shape:** The EOL provides insights into the required shape of localization functions. Correlations of different variable pairs require differently shaped localization functions. Localization functions should not necessarily be symmetric in $ln(p)$ as seen for wind. Furthermore, the optimal center of a distance-dependent localization can deviate from the reference level. For example, correlations of temperature and wind peak below the tropopause if the reference level is above the boundary layer. The maximum vertical correlation could indicate a suitable positioning of distance-dependent tapering functions. Finally, EOLs do not reveal a clear localization cut-off distance for tropospheric correlations. However, other considerations, e.g., continuity, computational efficiency or matrix rank, also may need to be considered when deciding on a cut-off.

– **Self- and cross-correlations:** Self- (e.g., temperature-temperature) and cross-correlations (e.g., temperature-humidity) should be localized differently. This fact could allow the development of correlation-dependent localization approaches. For example, self-correlations require no localization at zero distance, while the amplitude of cross-correlations should

be tapered by at least 25 %. Differently treating self- and cross-correlations resulted in performance close to a variable-dependent localization.

  – **Domain-uniform localization:** An tuned uniform distance-dependent localization using Gaspari-Cohn functions reduces the sampling error by about 10 %. Using tapering functions with an optimal shape could improve the localization substantially. The maximum error reduction was found for domain-uniform, variable, and height-dependent EOLs with
about 27 % improvement. Distinguishing between self- and cross-correlations leads to a similar but slightly smaller error reduction.

  – **Adaptive localization:** A statistical sampling error correction (SEC) achieves similar error reduction as a variable- and domain-uniform localization. Combining the SEC with a Gaspari-Cohn localization improves the error reduction. However, combining distance-dependent and statistical approaches requires re-tuning of localization scales. Combining
SEC and EOLs led to an over-correction of correlations, which slightly degraded the error reduction. This change could be related to sub-optimal prior assumptions when deriving SEC, as discussed by Anderson (2016) and Necker et al. (2020b).

Our results allow a better understanding of the requirements for vertical localization. When employing these conclusions, it is important to consider the specific demands of different ensemble filter algorithms. In ensemble transform Kalman filters,
localization increases the degrees of freedom of the analysis and thereby enables the assimilation of more observations (Hotta and Ota, 2021). Furthermore, our evaluation excluded considerations about the rank of the error covariance matrix and computational efficiency. Hence, our findings might need to be adapted to improve the analysis performance depending on the data assimilation system. Localization in operational NWP has many system-dependent requirements and is tuned to avoid bad signal-to-noise ratios during assimilation. For example, while we find no strong support for a vertical cut-off within the
troposphere for some variables, this could be beneficial due to the reasons discussed above.

How to apply EOL estimates will vary with the data assimilation algorithm as the application of localization is highly algorithm-specific. In case of covariance localization, constructing a generally non-SPSD localization matrix based on the EOL does not guarantee a symmetric positive semi-definite localized covariance matrix. However, different approaches allow for achieving positive semi-definiteness of localization matrices. Applying an NCM algorithm (Higham, 2002) for achieving
positive semi-definiteness resulted in only very minor changes of the EOL that hardly affect the error reduction.

For a serial filter (e.g., the Ensemble Adjustment Kalman Filter (EAKF) by Anderson (2001)), an EOL-based localization can be applied directly and it is planned to test this in follow-on studies. The EAKF does not involve a Schur product localization of a covariance matrix as each single observation is assimilated at a time, serially. Instead, the EAKF localizes the increment or gain. The gain between the observation and each state variable is multiplied by a scalar between 0 and 1. This localization
factor can be provided by the EOL.

Our study solely judges localization based on ensemble sampling error, assuming the 1000-member ensemble correlation as truth. It is difficult to predict the number of ensembles needed to apply our method, as it will vary for differing scenarios. However, we do not expect our results to change drastically if we had a larger ensemble. Besides, it would be interesting

to compare the EOL with the ELF or GGF approach. For example, comparing ELF and EOL could allow to investigate other error sources in the assimilation that can influence localization (Anderson and Lei, 2013). However, a proper comparison would require an OSSE with a sufficiently large ensemble.

We have found robust results for a mid-latitude convective summer period. The ever-increasing computational capabilities will enable extended data sets and a higher vertical resolution that is comparatively coarse in the current setup. Furthermore, our approach can be easily applied to other large ensemble simulations to study additional aspects, including horizontal localization. Extending this analysis is desirable given that localization can depend on the underlying weather condition (Lei et al., 2015; Destouches et al., 2021). For example, using a global simulation with a higher model top would allow studying different geographical regions, seasons, and stratospheric correlations that are particularly important for satellite data assimilation (Lei et al., 2018; Scheck et al., 2020).

*Code and data availability.* Code and processed data such as derived empirical optimal localizations are shared on zenodo: (Necker, 2022). The 1000-member ensemble data-set and derived covariances and correlations (approximately 60 TB of data) are too large for an upload but available upon request.

*Author contributions.* Tobias Necker: Conceptualization, Methodology, Software, Data curation, Visualization, Writing – original draft preparation; David Hinger: Methodology, Writing – review and editing; Philipp Johannes Griewank: Methodology, Writing – review and editing; Takemasa Miyoshi: Resources, Writing – review and editing; Martin Weissmann: Conceptualization; Methodology; Writing – original draft preparation

*Competing interests.* Co-author Takemasa Miyoshi is editor at the copernicus journal Nonlinear Processes in Geophysics (NPG). However, the peer-review process was guided by an independent editor. The authors have no other competing interests to declare.

*Acknowledgements.* Many thanks to Juan Ruiz, Jago Silberbauer, Jeffrey Anderson, and other colleagues at the University of Vienna, RIKEN RCC-S in Kobe, or the LMU in Munich who contributed to this research. Furthermore, we want to thank the two reviewers and the editor for their helpful comments that allowed us to improve the manuscript. The open source project and Python package "xarray" ((Hoyer and Hamman, 2017)) has been used to process ensemble data.

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
