# Peer review of "Guidance on how to improve vertical covariance localization based on a 1000-member ensemble"

_EGUsphere, 2022_

## Referee Comment (RC2)

**Review of the manuscript egusphere-2022-434 "Guidance on how to improve vertical covariance localization based on a 1000-member ensemble" by Tobias Necker, David Hinger, Philipp Johannes Griewank, Takemasa Miyoshi, and Martin Weissmann.**

Pavel Sakov

August 8, 2022

**1    General comments**

The the manuscript presents a study conducted in a straightforward way. It considers a set of 1000-member ensembles as representing the true state error covariance, and then investigates how various approaches to the vertical localisation can minimise the vertical correlation errors in 40-member sub-ensembles. This line of research is coherent with the previous efforts of the authors in the atmospheric ensemble DA.

In my opinion, for what it is, the study is done in a methodical and comprehensive way, and provides helpful material for further studies in that direction. However, the manuscript provokes a few questions in a more general context.

**2    Questions**

There are two main questions to the study for me: (1) how rigorous is the adopted methodology, and (2) how relevant are the results for other geophysical EnKF systems.

1. **On the concept of statistical ensemble.**

The underlying assumption employed in the study is that the EnKF ensemble is a statistical ensemble, i.e. that it is composed of members drawn from the same pool. While this can be true to some degree for some EnKF systems, it also can be demonstrated to be wrong for other systems. The alternative view is that the EnKF ensemble is a unit carrying the state of the DA system, and that ensembles of different size can have rather different statistical properties. For example, it is possible that a 40-member sub-ensemble of a 1000-member EnKF will have qualitatively different correlation errors to an ensemble of a properly set 40-member EnKF.

This real or potential concern could be partly overcome by experimental testing of results with 40-member systems. I say "partly" here because these experiments would still be conducted in a very specific environment.

2. **On the importance of the "right" localisation.**

   While localisation is a necessary attribute of large-scale EnKF systems, the sensitivity of the performance to the details of its implementation can be rather flat. From our experience with global ocean EnKF forecasting systems increasing or decreasing the horizontal localisation radius by say factor of 1.5 results to marginal changes in forecast innovation statistics. (Provided that the observation error variance is scaled proportionally to the localisation radius squared to keep the observation impact at the same level.)

   Therefore, I would suggest, firstly, to moderate claims of the importance of the choice of localisation technique for the forecasting skill of EnKF systems; and secondly, experimentally demonstrate the impact of the proposed taper functions.

**3   Conclusion**

I reiterate that in my view the study is conducted in a methodical and comprehensive way and would be interesting to specialists working on further advancements in that direction.

In a wider context, there remain grounds for scepticism in regard to the rigoursness of the underlying assumptions and applicability of the results to other systems. It seems to me that the study could benefit from experimental testing of the results. Also, it would be interesting to get some insight on implementation of the vertical localisation in the LETKF systems used.

I recommend to **accept** the paper for publication in NPG.

---

## Editor Comment (EC1)

Following the comments of the two referees, I would like as Editor to raise a point that I think is important, and possibly critical for acceptance of the paper. It is the symmetric positive semi-definite character of the matrices that are defined in the paper for representing localized covariances and correlations.

As reminded by the authors, a covariance matrix (and in particular a correlation matrix) must be symmetric positive semi-definite (SPSD, meaning without negative eigenvalues). If that condition is not verified in an EnKF, the minimization of the variance of the estimation error that is implicit in the analysis step of the EnKF (as also in variational assimilation) may lead to negative 'minima' (actually saddle-points) and to absurd results.

I understand that all localized correlations determined in the paper are obtained through formulas of the type of Eq. (3), *i. e.*, through Schur-multiplication of an original covariance matrix $\mathbf{P}$ by a localization matrix $\mathbf{C}$. For a given localization matrix $\mathbf{C}$, the localized matrix $\mathbf{P}_{loc}$ defined by the Schur-product will be SPSD for any covariance matrix $\mathbf{P}$, if, and only if, the localization matrix is itself SPSD.

Is the condition that the obtained matrices are SPSD verified in the paper ? As a precise example, do the quantities obtained through the EOL minimization Eq. (5-7) define (as it seems to be the authors' purpose) a correlation matrix, *i.e.* an SPSD matrix with 1's on its diagonal (the remark ((ll. 374-375) … *the EOL exhibited values larger than one when estimated after applying the SEC* suggests that all 'correlations' defined in the paper are not proper correlations) ?

These questions are not discussed, nor even mentioned, in the paper. I think they should be. It is possible that they have been discussed in previous papers (either by the authors of the present paper or by other authors), where responses that are relevant for the present paper can be found. If so, appropriate references and explanations must be given.

If not, I think it is necessary to check the SPSD character of either the localization matrix $\mathbf{C}$ or the localized matrix $\mathbf{P}_{loc}$ (or both). That can be done on the basis of theoretical considerations (it is not clear to me if the correlation matrices defined by the EOL minimization Eq. (5-7) are even symmetric). Or it can be done alternatively through numerical computations. There are in the present case 4 physical variables and 20 vertical levels, so that the relevant matrices have dimension 80 x 80, of which it must be possible to determine explicitly their full spectrum of eigenvalues. And if that is too costly, it is possible to consider submatrices, for instance by reducing the number of vertical levels.

If matrices that are meant to be SPSD, while being symmetric, turn out not to be SPSD, but with only a small number of small negative eigenvalues, one solution may be to set those eigenvalues to 0 (or to small positive values). If the negative character of the matrices is significant, there will be a real problem, which will have to be solved or at least discussed in depth. It may be that the conclusion of the paper will be that difficulties remain, to be solved in future works.

In any case, I consider as Editor that a proper discussion of those SPSD aspects is critical for acceptance of the paper.

I add one remark. The authors mention on several occasions (*e.g.* ll. 44-45) distance-dependent tapering functions with a cut-off at finite distance. A distance-dependent SPSD function that is continuous at the origin (*i.e.* at distance 0) cannot have a discontinuity

elsewhere (that would be inconsistent with the requirement that the correlation between two close points must tend to 1 when the distance between those points tends to 0). It may be that people who have used such 'cut-off' functions have not run into difficulties because of the 'small' negativity of the corresponding covariances-correlations, but those functions cannot mathematically be SPSD.

---

## Editor Comment (EC2)

To the authors,

I thank you for the revised version of your paper. You now discuss the symmetric positive semi-definite (SPSD) character of the localization matrix $\mathbf{C}$. You also mention in the response in the Interactive Discussion that this matrix is not SPSD (with eigenvalues ranging from -0.5 to 50, which suggests that the negative part of the matrix is in a sense 'small'). You also mention that the matrix $\mathbf{C}$' obtained by setting the negative eigenvalues to 0 changes the localized values by up to 15%.

These results are interesting, and should in my mind be included (together with other possible results) in your paper. More generally, I think that the diagnostics you have performed on the basis of the matrix $\mathbf{C}$ (starting with subsection 3.1) should be performed again with the corrected matrix $\mathbf{C}$' (or actually with any other appropriate SPSD matrix you might build from your original matrix $\mathbf{C}$). I do not think the required cost would be prohibitive, and that would greatly increase the interest and significance of your paper. Actually, the diagnostics you have performed with your matrix $\mathbf{C}$ could be removed from the paper if they are replaced with new diagnostics obtained with an SPSD localization matrix. It may be that the new diagnostics will turn out to be similar to the previous ones. If so, that would actually be very instructive.

I make three additional remarks

- It would be appropriate to mention explicitly that the matrix $\mathbf{P}_{loc}$ defined by the Schur product (3) is SPSD for any SPSD matrix $\mathbf{P}$ if, and only if, the localization matrix $\mathbf{C}$ is itself SPSD (I cannot mention an explicit reference for that, although I know there exist some, but I can provide you with a proof if necessary).

- You write *As some DA algorithms require SPSD covariance matrices* (l. 237 of your Authors'.Tracked.changes file) and *covariances or localization matrices often need to be symmetric positive semi-definite* (l. 465). Well, covariance matrices are SPSD by definition. As for DA algorithms, all algorithms that are based, explicitly or implicitly, on a Bayesian approach (and that is the case of Kalman Filter and of all its variants, particularly ensemble variants, in which probability distributions are described by finite samples) require proper covariance matrices (if that condition is not verified, the corresponding computations are meaningless and can lead to absurd results, as has actually been observed by many authors).

- Localization is necessary in the various forms of Ensemble Kalman Filter because of the relatively small size of the ensembles that are used. It might be useful, if this is possible, to give some quantitative information on how the need for localization decreases when the ensemble size increases.

I am going to ask you to make a major revision of your paper along the lines above. I will submit your new revised version to further review.

---

## Author Response (AR1)

**Response to comments/reviews - *egusphere-2022-434*:**

**Guidance on how to improve vertical covariance localization
based on a 1000-member ensemble**

**by Tobias Necker, David Hinger, Philipp Johannes Griewank, Takemasa Miyoshi, and Martin Weissmann**

We would like to thank all reviewers and the editor for valuable comments and suggestions. This document combines all our responses into a single document/pdf. Below you can find the original comments by the reviewers followed by our response including explanations of required changes in the revised manuscript. **The original queries are bold/black**. Our responses are in normal text/blue. *Changes in the manuscript are highlighted italic.*

**RC1**: 'Comment on egusphere-2022-434', Lili Lei, 04 Jul 2022

**Summary**

**This manuscript uses a convection-permitting 1000-member ensemble simulation to examine the vertical localization. An empirical optimal localization is proposed, which minimizes the sampling error of correlations estimated from a 40-member ensemble comparing to those from the 1000-member ensemble. Vertical correlations and localization functions for different state variable and cross variables are systematically examined. Results show that different vertical localization functions are required for different variables and vertical height. Combination of the empirical optimal localization with adaptive sampling error correction is also investigated. The manuscript is well-written and could be a valuable contribution for the data assimilation community. I have several comments as below.**

**1) l22, It is more appropriate to say members of O(100), since Canadian Center has 2 groups of 128-member ensembles.**
Reply: We are aware that Environment Canada has a larger ensemble than other forecasting centres and tried to address this by using the word "*usually*". To our knowledge Environment Canada is the only centre that exceeds 100 members, while most other centres use about 40-50 members. For this reason, we think that O(100) might be misleading.
Some ensemble sizes:
En. Ca. - EnKF - 256 members
NCEP - deterministic EnSRF - 80 members
ECMWF - hybrid 4DVAR - 50 members
JAM - hybrid LETKF/4D-Var- 50 members
MeteoFrance - hybrid 4DVAR - 50 members
Korea - KIAPS-LETKF - 50 members
DWD - LETKF - 40 members

**2) l100-105, the description of BCs is confusing. Is the GEFS 20-member analysis used 50 times to get 1000 BCs? Or climatological GEFS is sampled for 1000 BCs?**
Reply: The 20-member analysis ensemble is used 50 times to reach 1000 BCs and afterwards combined with 1000 random climatologically scaled perturbations.
To avoid confusion, we will change the following sentences: "*These BCs combine 1000 climatologically scaled random perturbations with an analysis ensemble of the NCEP Global Ensemble Forecast System (GEFS). The GEFS 20-member analysis ensemble is used 50 times to reach 1000 BCs and afterwards combined with 1000 random climatologically scaled perturbations.*"

**3) l197, there are increased correlations below 800hPa. Are there any physical explanations for this?**

Reply: Following your question, we analysed temperature and hydrometeor profiles in the ensemble. Members with colder mid-tropospheric temperatures exhibited more upper-level clouds resulting in colder near-surface temperatures that likely are caused by stronger cloud shadowing.

We will add the following comment: "*This weak correlation is linked to cloud shadowing by mid-tropospheric clouds and resulting colder near-surface temperatures.*"

**4) Eq. (5), is s the index for 40-member groups with S=25? Eq. (5) is similar to Eq. (1) in Lei and Anderson (2014).**

Reply: Yes, the index for S is 25 as we use 25 40-member sub-samples (groups) to compute the EOL. We will add the following sentence: "*K is the number of vertical columns in the domain, and S is the number of subsamples.*"

Eq. (5) is similar to Eq. (2) in Anderson (2007) and to Eq. (1) in Lei and Anderson (2014) but exhibits two essential differences: On the one hand, we consider the correlation coefficient instead of the regression coefficient. On the other hand, our cost function minimises the sampling error with respect to the 1000-member truth, which also results in different sums.

**5) If the sample correlation $r_{40}$ tends to overestimate the true correlation $r_{1000}$, the EOL computed from Eq. (7) should be no larger than 1.0? As discussed by Lei and Anderson (2014), ELF can account for inflation compared to GGF. But I am not sure about the EOL in Eq. (7), which could be true since the true correlation is known. Can the authors provide some derivations on this statement?**

Reply: The EOL can inflate sample correlations similar to the ELF but only by optimising sample correlations. The EOL can reach values larger than 1.0 if the true correlation/$r_{1000}$ is larger than the sample correlation/$r_{40}$. In our setting, this is unlikely as we apply multiple sample correlations that are usually larger than the true correlation given a sufficient number of subsamples. However, we got EOL values larger than 1 when combining the SEC with the EOL (see, for example, Fig 2 in the Appendix below). The EOL inflated sample correlations when the SEC was applied first and damped sample correlations too strong.

We will add the following sentences: "*Values larger than one can occur when the true correlation is larger than the sample correlation. For example, this can happen when estimating the EOL after applying other localization approaches.*"

**6) Figs 2 and 7, how about the sample correlations estimated for cross variables?**

Reply: It would also be possible to show and discuss the sample correlations. However, we believe that providing the sample correlations adds little additional information that is not already supplied by the curves in Fig 2 and the corresponding EOL in Fig 3 (Fig 7 and 8, respectively). We. therefore, believe that including extra lines or figures would rather distract the reader. Fig 3 in the Appendix below shows the sample correlations for single variable pairs and 500hPa.

**7) Figs 3-5, the UU EOL seems have values larger than 1.0. What is the exact value at the reference level? Why EOL estimates localization larger than 1.0 when sample correlations are close to 1.0? Intuitively, when sample correlations are close to the true correlations as 1.0, localization value goes to 1.**

Reply: We modified the x-axis (extended x range) in Figs 3-5, so the reader can see that the EOL values do not surpass 1.0 (see, for example, Fig 1 in the Appendix below). At the reference level, the EOL reaches a value of 1.0. Please also refer to our reply to comment 5, which addresses a similar point.

**8) l258-260, this discussion is based on sampling errors in correlations. But for cycling data assimilation experiments, too strong taper for cross variables may result in too weak corrections.**

Reply: Unfortunately, based on our experimental setting, we can only judge sampling errors in background sample correlations, which excludes a cycled assimilation environment. This means that the EOL is optimal in terms of sampling error in correlations but not necessarily optimal in terms of analysis or cycling performance.

**9) l290, the "error reduction" is for estimated correlation, not for prior/posterior errors by using the EOL. Also it would be helpful to have some discussions about the estimated localization and localization applied for cycling data assimilation in the section of conclusions and discussions.**

Reply: As mentioned in the previous point, we only can estimate localization and error reduction based on the background correlation/covariance. We want to avoid speculation and therefore prefer not to discuss potential localisation changes that might impact the error in a single or cycled analysis. For clarity, we will add the following sentence: "*The result can be interpreted as a benchmark of the maximum possible correlation error reduction achieved by a domain-uniform height and variable-dependent localization. Note that results for optimizing the analysis may lead to different optimal localization values under some circumstances, but this is beyond the scope of this paper.*"

**10) Section 3.1, a curious question, if the direct-variable EOL is applied rather than cross-variable EOLs, i.e., UU is applied for UU, UV, UT, and UQ, how about the correlation error reduction?**

Reply: This is an interesting question. We tested this approach in one experiment and found an error reduction similar to ALL or SEC. We plan to add the following sentences: "*Additionally, we tested the error reduction for applying the EOL estimated for self-correlations to both self- and cross-correlations of each variable (e.g., EOL derived from TT applied to TT, TQ, TU, and TV). For this setting, the error reduction was similar to ALL or SEC (not shown), which underlines the need to treat self- and cross-correlations differently.*"

**11) l396-399, it would be helpful to add some dynamical explanations for these results.**

Reply: Finding dynamical explanations would be interesting. We attempted to incorporate such a dynamical interpretation, but it is hard to distinguish between a coincidental correlation or a dynamically induced correlation (correlation does not imply causation). Additionally, we show results averaged over a very large sample of profiles during different atmospheric conditions. Thus, it seems impossible to provide a short explanation of dynamical reasons without speculation.

**12) l407, the computational efficiency issue can be treated by model-space localization (Lei et al. 2018).**

Reply: Keeping in mind the full range of ensemble DA methods implemented in various places, we are convinced that it is generally correct to say that other factors, e.g. computational efficiency, may also need to be considered. However, we will add "*may*" in this sentence to weaken the statement.

Fig 1 Extended x-range

[Figure]

Fig 2: Only SEC + EOL shows EOL values larger than 1

[Figure]

Fig 3 Sample correlations

[Figure]

Fig 4 True correlations for comparison with Fig 3 (Note: This is Fig 2 from the manuscript)

[Figure]

**RC2: 'Comment on egusphere-2022-434'**, Pavel Sakov, 08 Aug 2022

**Review of the manuscript egusphere-2022-434**

**"Guidance on how to improve vertical covariance localization based on a 1000-member ensemble" by Tobias Necker, David Hinger, Philipp Johannes Griewank, Takemasa Miyoshi, and Martin Weissmann.**

**Pavel Sakov August 8, 2022**

**1 General comments**

The manuscript presents a study conducted in a straightforward way. It considers a set of 1000-member ensembles as representing the true state error covariance, and then investigates how various approaches to the vertical localisation can minimise the vertical correlation errors in 40-member subensembles. This line of research is coherent with the previous efforts of the authors in the atmospheric ensemble DA.

In my opinion, for what it is, the study is done in a methodical and comprehensive way, and provides helpful material for further studies in that direction. However, the manuscript provokes a few questions in a more general context.

**2 Questions**

There are two main questions to the study for me: (1) how rigorous is the adopted methodology, and (2) how relevant are the results for other geophysical EnKF systems.

**1. On the concept of statistical ensemble.**

The underlying assumption employed in the study is that the EnKF ensemble is a statistical ensemble, i.e. that it is composed of members drawn from the same pool. While this can be true to some degree for some EnKF systems, it also can be demonstrated to be wrong for other systems. The alternative view is that the EnKF ensemble is a unit carrying the state of the DA system, and that ensembles of different size can have rather different statistical properties. For example, it is possible that a 40-member sub-ensemble of a 1000-member EnKF will have qualitatively different correlation errors to an ensemble of a properly set 40-member EnKF.

This real or potential concern could be partly overcome by experimental testing of results with 40-member systems. I say "partly" here because these experiments would still be conducted in a very specific environment.

Reply: Thanks for raising this point. Indeed, we assume that our EnKF ensemble is a statistical ensemble without mentioning it explicitly. Please also keep in mind that our convective ensemble is initialized from a downscaled analysis, which might impact the statistics, too.

Unfortunately, testing this assumption is beyond the scope of the present study, but we will consider investigating this assumption in future research. For clarity, we will add the following sentence in Sec. 2.4: "*We assume that the 40-member sub-ensembles of the 1000-member ensemble statically have sampling errors similar to those independent 40-member ensemble EnKF systems would have.*"

**2. On the importance of the "right" localisation.**

**While localisation is a necessary attribute of large-scale EnKF systems, the sensitivity of the performance to the details of its implementation can be rather flat. From our experience with global ocean EnKF forecasting systems increasing or decreasing the horizontal localisation radius by say factor of 1.5 results to marginal changes in forecast innovation statistics. (Provided that the observation error variance is scaled proportionally to the localisation radius squared to keep the observation impact at the same level.) Therefore, I would suggest, firstly, to moderate claims of the importance of the choice of localisation technique for the forecasting skill of EnKF systems; and secondly, experimentally demonstrate the impact of the proposed taper functions.**

Reply: Thank you for providing insights on your practical experience regarding localization in the ocean EnKF system. We are currently working on demonstrating the impact of EOL-based localization functions in an OSSE. These ongoing experiments will provide further insights and likely be part of a future manuscript. Especially for satellite DA, we already see a large influence of vertical localization.

Our present analysis only allows evaluating localization and error reduction based on the background error correlation judged by the 1000-member ensemble truth. We want to avoid speculation and therefore do not discuss the potential impact the choice of localisation might have on a single or cycled analysis (or forecast).

For clarity, we will add the following sentence: "*The result can be interpreted as a benchmark of the maximum possible correlation error reduction achieved by a domain-uniform height and variable-dependent localization. Note that results for optimizing the analysis may lead to different optimal localization values under some circumstances, but this is beyond the scope of this paper.*"

Furthermore, we will weaken the statement in line 71 ("is crucial" -> "has the potential"): "*Consequently, a better understanding of optimal vertical localization for convection-permitting simulations has the potential to improve forecasts of convective precipitation and related hazards.*"

**3 Conclusion**

**I reiterate that in my view the study is conducted in a methodical and comprehensive way and would be interesting to specialists working on further advancements in that direction.**

**In a wider context, there remain grounds for scepticism in regard to the rigoursness of the underlying assumptions and applicability of the results to other systems. It seems to me that the study could benefit from experimental testing of the results.**

Also, it would be interesting to get some insight on implementation of the vertical localisation in the LETKF systems used.

I recommend to accept the paper for publication in NPG.

**EC1:** **'Comment on egusphere-2022-434'**, Olivier Talagrand, 17 Aug 2022

Following the comments of the two referees, I would like as Editor to raise a point that I think is important, and possibly critical for acceptance of the paper. It is the symmetric positive semi-definite character of the matrices that are defined in the paper for representing localized covariances and correlations.

As reminded by the authors, a covariance matrix (and in particular a correlation matrix) must be symmetric positive semi-definite (SPSD, meaning without negative eigenvalues). If that condition is not verified in an EnKF, the minimization of the variance of the estimation error that is implicit in the analysis step of the EnKF (as also in variational assimilation) may lead to negative 'minima' (actually saddle-points) and to absurd results.

I understand that all localized correlations determined in the paper are obtained through formulas of the type of Eq. (3), i. e., through Schur-multiplication of an original covariance matrix P by a localization matrix C. For a given localization matrix C, the localized matrix Ploc defined by the Schur-product will be SPSD for any covariance matrix P, if, and only if, the localization matrix is itself SPSD.

Is the condition that the obtained matrices are SPSD verified in the paper? As a precise example, do the quantities obtained through the EOL minimization Eq. (5-7) define (as it seems to be the authors' purpose) a correlation matrix, i.e. an SPSD matrix with 1's on its diagonal (the remark ((ll. 374-375) … the EOL exhibited values larger than one when estimated after applying the SEC suggests that all 'correlations' defined in the paper are not proper correlations)?

These questions are not discussed, nor even mentioned, in the paper. I think they should be. It is possible that they have been discussed in previous papers (either by the authors of the present paper or by other authors), where responses that are relevant for the present paper can be found. If so, appropriate references and explanations must be given.

If not, I think it is necessary to check the SPSD character of either the localization matrix C or the localized matrix Ploc (or both). That can be done on the basis of theoretical considerations (it is not clear to me if the correlation matrices defined by the EOL minimization Eq. (5-7) are even symmetric). Or it can be done alternatively through numerical computations. There are in the present case 4 physical variables and 20 vertical levels, so that the relevant matrices have dimension 80 x 80, of which it must be possible to determine explicitly their full spectrum of eigenvalues. And if that is too costly, it is possible to consider submatrices, for instance by reducing the number of vertical levels.

If matrices that are meant to be SPSD, while being symmetric, turn out not to be SPSD, but with only a small number of small negative eigenvalues, one solution may be to set those eigenvalues to 0 (or to small positive values). If the negative character of the matrices is significant, there will be a real problem, which will have to be solved or at least discussed in depth. It may be that the conclusion of the paper will be that difficulties remain, to be solved in future works.

**In any case, I consider as Editor that a proper discussion of those SPSD aspects is critical for acceptance of the paper.**

Reply: Thank you for raising this important point. Overall, one central aim of our study was to analyse how an optimal vertical localization should look like without given algorithmic constraints. The SPSD requirement is one of those constraints. We agree that it would be helpful to discuss this aspect to make the reader and future studies aware of this constraint. Following your comment, we performed some additional analysis in this regard, which we discuss below. Furthermore, we will discuss the SPSD aspect in the revised manuscript, given its importance for EnKF and variational systems.

The localization matrix C based on the (SINGLE) EOL is symmetric by definition given Eq. 7. The diagonal contains all 1's. As suggested, we analysed the C matrix with dimensions 80x80 (see the Appendix/Supplement, Figure 1). In this example, the resulting localization matrix was symmetric but not PSD. The eigenvalues of this C matrix range from -0.5 to 50, while more than half of the eigenvalues are positive. Setting all negative eigenvalues to zero and constructing a C' matrix that is SPSD (see Figure 2) changes the localization values by up to 15% (see Figure 3).

We will add a discussion in Sec. 2.5: *"Applying the EOL by construction yields a symmetric but not necessarily a positive semi-definite localization matrix. In our case, the computed localization matrices are not symmetric positive semi-definite (SPSD), which can result in non-SPSD localized covariance matrices. As some DA algorithms require an SPSD covariance matrix (Gaspari and Cohn 1999, Bannister 2008), additional steps would be required to apply the EOL results to such algorithms."*

We will add a discussion in the *conclusion section: "For a serial filter (e.g., the Ensemble Adjustment Kalman Filter (EAKF) by Anderson2001), an EOL-based localization can be applied directly and easily tested in future studies. Yet, each filter can exhibit algorithm-specific requirements for localization. For example, covariances or localization matrices often need to be symmetric positive semi-definite, which the EOL methodology might not fulfil. However, in all cases, EOL results can serve as guidance for finding better localization functions or methods that resemble the results of the EOL but also fulfil the criteria of a symmetric positive semi-definite matrix."*

**"(the remark ((ll. 374-375) … the EOL exhibited values larger than one when estimated after applying the SEC suggests that all 'correlations' defined in the paper are not proper correlations)?"**

Reply: In our study, correlations are proper correlations. EOL values larger than one occur when the SEC was previously applied to statistically correct sampling error in correlations. Yet, correlations sometimes are damped too much due to the suboptimal behaviour of the SEC. In this case, the EOL allows the diagnosis of deficiencies in the applied localization approach. EOL values larger than one indicate that the SEC damps correlations too much, while EOL values smaller than one reveal that the SEC did not successfully correct sampling errors.

**I add one remark. The authors mention on several occasions (e.g. ll. 44-45) distancedependent tapering functions with a cut-off at finite distance. A distance-**

**dependent SPSD function that is continuous at the origin (i.e. at distance 0) cannot have a discontinuity elsewhere (that would be inconsistent with the requirement that the correlation between two close points must tend to 1 when the distance between those points tends to 0). It may be that people who have used such 'cut-off' functions have not run into difficulties because of the 'small' negativity of the corresponding covariances-correlations, but those functions cannot mathematically be SPSD.**

Reply: Thanks for this interesting remark. Indeed, a cut-off could lead to discontinuities. However, our paper only applies and refers to the Gaspari-Cohn (GC) function when discussing cut-offs. The GC function is a piecewise rational function which is continuous (Gaspari and Cohn, 1999). This property explains why people do not run into difficulties caused by negativity when using the GC function that exhibits a cut-off (damps correlations to zero after a defined distance).

We changed the following sentences to be more precise and to make people aware of the importance of continuity:

*"Distance-dependent localization always requires tuning of localization scales and cut-off distances (deleted)."*

*"This behaviour motivates most distance-based localization approaches with a predefined tapering function that damp or cut-off (deleted) distant correlations."*

*"However, other considerations, e.g.,* **continuity***, computational efficiency or matrix rank, also may need to be considered when deciding on a cut-off."*

Figure 1: Localization matrix based on domain-uniform EOL (SINGLE variable case)

[Figure]

Figure 2: SPSD localization matrix constructed using non-negative eigenvalues

[Figure]

Figure 3: Difference between matrix C and C' showed in Figures 1 & 2

[Figure]

---

## Author Response (AR2)

**Response to comment/review -** *egusphere-2022-434***:**

**Guidance on how to improve vertical covariance localization
based on a 1000-member ensemble**

**by Tobias Necker, David Hinger, Philipp Johannes Griewank, Takemasa Miyoshi, and
Martin Weissmann**

We would like to thank the editor for his helpful comments and for answering questions.
During the revision process, we performed additional computations and studied different
approaches that allow for achieving positive semi-definiteness of a localization matrix. The
revised version of the manuscript includes these outcomes, which we summarize in a newly
added section. We also expanded the theory and conclusion sections, which now discuss
the positive semi-definite character of covariances and localization.
Below we respond to the editor's comments and further explain how we addressed them in
the revised manuscript. The original queries are bold, and our responses are normal text. All
changes in the manuscript are clearly marked in the pdf latex diff document.

**EC2: 'Reply on AC3', Olivier Talagrand, 19 Sep 2022**

**To the authors,**

**I thank you for the revised version of your paper. You now discuss the symmetric
positive semi-definite (SPSD) character of the localization matrix C. You also mention
in the response in the Interactive Discussion that this matrix is not SPSD (with
eigenvalues ranging from -0.5 to 50, which suggests that the negative part of the
matrix is in a sense 'small'). You also mention that the matrix C' obtained by setting
the negative eigenvalues to 0 changes the localized values by up to 15%.**

**These results are interesting, and should in my mind be included (together with other
possible results) in your paper. More generally, I think that the diagnostics you have
performed on the basis of the matrix C (starting with subsection 3.1) should be
performed again with the corrected matrix C' (or actually with any other appropriate
SPSD matrix you might build from your original matrix C). I do not think the required
cost would be prohibitive, and that would greatly increase the interest and
significance of your paper. Actually, the diagnostics you have performed with your
matrix C could be removed from the paper if they are replaced with new diagnostics
obtained with an SPSD localization matrix. It may be that the new diagnostics will turn
out to be similar to the previous ones. If so, that would actually be very instructive.**

Answer: We addressed your comments and substantially revised the manuscript with
respect to the SPSD aspect. We now compare different approaches that allow for achieving
SPSD of the EOL localization matrix. This analysis includes the application of the nearest
correlation matrix (NCM) algorithm (Higham 2002), which is a useful tool and, to our
knowledge, was never applied in a data assimilation context.
Overall, we found that guaranteeing SPSD of the constructed localization matrix results in
minor changes in the raw EOL estimate (see Fig. 1&2 in the Appendix). The conclusions of
the error reduction are unaffected when using an SPSD localization. For this reason, we
mainly present "raw" EOL results that are optimal in terms of sampling error reduction
without any additional constraints.

The revised version of the manuscript includes additional results, which we summarize in a newly added section. The new section analyzes changes due to enforcing SPSD by discussing Figure 1 (see Appendix/Supplement). We also adapted the theory and conclusion sections, which now discuss the SPSD characteristic of covariances and localization in more depth.

**I make three additional remarks**

**- It would be appropriate to mention explicitly that the matrix Ploc defined by the Schur product (3) is SPSD for any SPSD matrix P if, and only if, the localization matrix C is itself SPSD (I cannot mention an explicit reference for that, although I know there exist some, but I can provide you with a proof if necessary).**

Answer: Done. We now reference the Schur product theorem and adapted the theory section accordingly.

**- You write As some DA algorithms require SPSD covariance matrices (l. 237 of your Authors'.Tracked.changes file) and covariances or localization matrices often need to be symmetric positive semi-definite (l. 465). Well, covariance matrices are SPSD by definition. As for DA algorithms, all algorithms that are based, explicitly or implicitly, on a Bayesian approach (and that is the case of Kalman Filter and of all its variants, particularly ensemble variants, in which finite samples describe probability distributions) require proper covariance matrices (if that condition is not verified, the corresponding computations are meaningless and can lead to absurd results, as has actually been observed by many authors).**

Answer: Indeed, this sentence was not well phrased, as all covariance matrices should be symmetric positive semi-definite. We adopted the conclusion based on our latest results. Overall, we agree that proper mathematical covariances are a crucial ingredient for data assimilation. However, we struggled to find literature that describes how the negative definiteness of covariance matrices or localization affects the estimation process. To avoid speculation, we decided not to discuss the effect of definiteness on the estimation process.

**- Localization is necessary in the various forms of Ensemble Kalman Filter because of the relatively small size of the ensembles that are used. It might be useful, if this is possible, to give some quantitative information on how the need for localization decreases when the ensemble size increases.**

Answer: Thank you for this comment. We have already thought about studying the effect of ensemble size on localization and consider to do so in the future. However, such an analysis is beyond the scope of the present study as it comprises a new question.

**I am going to ask you to make a major revision of your paper along the lines above. I will submit your new revised version to further review.**

_References:_ _Nicholas J. Higham, Computing the nearest correlation matrix—a problem from finance, IMA Journal of Numerical Analysis, Volume 22, Issue 3, July 2002, Pages 329–343,_ _https://doi.org/10.1093/imanum/22.3.329_

*Appendix/Supplement*

Figure 1: Examples of EOL-based localization matrices for a single vertical column: (a) matrix C constructed based on the \textit{SINGLE} case, (b) resulting nearest correlation matrix C following the NCM algorithm, and (c) changes due to enforcing positive definiteness. (This figure is now included in the manuscript as Fig. 10.)

[Figure]

Figure 2: Similar data/analysis as in Figure 1 but snapshots of the matrices showing temperature and reference level 500hPa.
a) EOL          b) SPSD EOL          c) Differences

---

## Editor Decision (ED2)

As you may have seen, one referee has submitted a comment on the new revised version of the paper. Although she has not let her name explicitly known this time, the referee is Lili Lei, who was referee 1 of the first version of your paper (she has agreed that I let again her name known to you).

She only asks you to comment on the link between inflation and the SPSD character of localization matrices. Please respond to that comment.

As Editor, I also have a number of comments that you will find below (the line numbers are those of file egusphere-2022-434-ATC2.pdf, which contains explicitly the modifications you have made on the previous version).

You have now studied in some detail the SPSD character of the localized matrices you have obtained. I have first a few remarks on this point.

1. The need for localized matrices to be SPSD should be mentioned, and at least briefly discussed, in your introduction (and mentioned in particular in the paragraph ll. 85-91 where you present the general structure your paper).

2. You have used the nearest correlation matrix (NCM) algorithm defined by Higham (2002), which identifies the SPSD matrix that is nearest to a given symmetric matrix. It could be useful to mention by which measure the distance between two matrices is defined in that algorithm.

3. LL. 435-436, … *the SPSD localization performs only marginally worse than the EOL localization*. Obscure. From what I understand, there are no two localization methods to be compared. You have implemented the NMC algorithm on the results of EOL localization, and the result is a slight degradation of the error against the 1000-member ensemble.

4. Ll. 498-499. *Applying an NCM algorithm for achieving positive semi-definiteness resulted in only very minor changes of the EOL that hardly affect the error reduction*. Since that seems to be your main conclusion concerning the SPSD character of the localization matrix, I think it should be mentioned in the abstract. I add that, since EOL is intended at minimizing an error, it is not surprising that NCM increases that error. A more significant test (possibly for a future work) would be of course to consider the impact on the analysis which follows the localization.

And for other comments

5. Ll. 45-46 (and elsewhere) …. *a Gaussian-shaped tapering function* …. The words *Gaussian-shaped*, where I understand you actually mean *unimodal* (a unique maximum, with possibly symmetry about that maximum) is not very appropriate (in particular, the functions you consider become zero at finite distance).

6. L. 156, … *Gaspari-Cohn function (GC; Eq. 4.10, Gaspari and Cohn, 1999)* … It could be good to give the explicit expression of the function that you are referring to here.

7. L. 116. Mention the year (2016, according to the caption of several Figures)

8. L. 131, … *a sample of state vectors $x^n$* … (with explicit superscripts)

9. L. 186, … *with unique members …*, obscure. I understand *without overlap*.

10. L. 224, Eq. (5), mention that $S$ is the number of 40-member ensembles ($S = 25$)

11. L. 271, *The variability between day to night also appears to be small*. That does not seem to be visible from what you show. Insert the words *(not shown)* or something similar.
The same remark applies to other places, for instance to l. 357.

12. L. 366, what are the *GC setups* that are supposed to be listed in Sec. 2.3.1 ? Be more explicit, or remove.

13. L. 397, … *derogates the error reduction*. That makes no sense to me. Do you mean … *increases the error*. ?

14. Ll. 464-465, *The shapes that we found included Gaussian, exponential or linear functions*. Well, you have shown none of that.

15. Fig. 7. The standard deviations on the *All* curves cannot be seen (whether on the screen or on a print-out). If standard deviations are so small that they cannot be distinguished from the (rather thick) curves themselves, say it (the same remark may apply to other figures, please check that aspect carefully).

16. L. 448, *Our analysis includes three prognostic variables (humidity, temperature, and wind)*. Well, you have so far spoken of four variables, including the two wind components.

Please revise your paper according to Lili Lei's comment, as well as to mine. In case you disagree with a particular comment, or decide not to follow a particular suggestion, please state precisely your reasons for that.

I thank you for having thought of *Nonlinear Processes in Geophysics* for submitting your paper, and look forward to receiving a further revised version.

REFERENCE

Higham, N. J.: Computing the nearest correlation matrix—a problem from finance, IMA Journal of Numerical Analysis, 22, 329–343, https://doi.org/10.1093/imanum/22.3.329, 2002.

---

## Author Response (AR3)

**Response to minor review/comments - *egusphere-2022-434*:**

**Guidance on how to improve vertical covariance localization
based on a 1000-member ensemble**

**by Tobias Necker, David Hinger, Philipp Johannes Griewank, Takemasa Miyoshi, and Martin Weissmann**

We would like to thank the editor and reviewer for their additional comments. We appreciate the time invested during the revision process. Below we respond to the comments and further explain how we addressed them in the revised manuscript. The **original queries are bold**, and our responses are normal text. We explain all changes and provided line numbers (e.g., L1 = Line 1) that refer to the newly submitted pdf latex diff document.

**RC3**
**Suggestions for revision by Submitted on 11 Nov 2022**
**Anonymous referee #1**

**Discussions about the symmetric, positive semi-definite (SPSD) feature for the localization matrix are added. Several potential ways to guarantee a SPSD localization matrix are mentioned. As the authors mentioned in the text that EOL can play the role of inflation or deflation if needed. It would be helpful to add a little discussion about the impact of inflation on the SPSD localization matrix.**
Indeed, the impact of inflation (of variances) on SPSD would be an interesting topic to study but appears beyond the scope of our paper.
Please note that the EOL does not play the role of "inflation," as it does not inflate or deflate variances. By definition, the EOL corrects sampling errors in correlations. Values larger or smaller than one on the diagonal of the localization matrix (which would impact variances) do not originate from the EOL but could be provoked by SPSD methods and, thus, should be fixed.
We removed or replaced the word "inflation" where required to avoid confusion for readers.
(L234; L400-401)

**EC3**
**As you may have seen, one referee has submitted a comment on the new revised version of the paper. Although she has not let her name explicitly known this time, the referee is Lili Lei, who was referee 1 of the first version of your paper (she has agreed that I let again her name known to you).**

**She only asks you to comment on the link between inflation and the SPSD character of localization matrices. Please respond to that comment.**
Please see our response to the reviewer's comment above.

**As Editor, I also have a number of comments that you will find below (the line numbers are those of file egusphere-2022-434-ATC2.pdf, which contains explicitly the modifications you have made on the previous version).**

**You have now studied in some detail the SPSD character of the localized matrices you have obtained. I have first a few remarks on this point.**
**1. The need for localized matrices to be SPSD should be mentioned, and at least briefly discussed, in your introduction (and mentioned in particular in the paragraph ll. 85-91 where you present the general structure of your paper).**
We now mention the need for SPSD localization in the introduction twice. (L77-78; L90-91)

**2. You have used the nearest correlation matrix (NCM) algorithm defined by Higham (2002), which identifies the SPSD matrix that is nearest to a given symmetric matrix. It could be useful to mention by which measure the distance between two matrices is defined in that algorithm.**
The distance is measured using weighted Frobenius norms. We added a comment to Sect. 3.4. in the manuscript. (L430-431)

**3. LL. 435-436, … the SPSD localization performs only marginally worse than the EOL localization. Obscure. From what I understand, there are no two localization methods to be compared. You have implemented the NMC algorithm on the results of EOL localization, and the result is a slight degradation of the error against the 1000-member ensemble.**
We rephrased the sentence to avoid confusion. (L437-438)

**4. Ll. 498-499. Applying an NCM algorithm for achieving positive semi-definiteness resulted in only very minor changes of the EOL that hardly affect the error reduction. Since that seems to be your main conclusion concerning the SPSD character of the localization matrix, I think it should be mentioned in the abstract. I add that, since EOL is intended at minimizing an error, it is not surprising that NCM increases that error. A more significant test (possibly for a future work) would be of course to consider the impact on the analysis which follows the localization.**
Indeed, it would be very important to investigate the impact on the analysis and forecast, which we hope to examine soon using an OSSE setup.
We added a short comment to the abstract and now mention the error reduction. A longer comment was not possible due to the word limit for the abstract (200 words). (L12)

**And for other comments**
**5. Ll. 45-46 (and elsewhere) …. a Gaussian-shaped tapering function …. The words Gaussian-shaped, where I understand you actually mean unimodal (a unique maximum, with possibly symmetry about that maximum) is not very appropriate (in particular, the functions you consider become zero at finite distance).**
We replaced Gaussian-shaped with Gaspari-Cohn, to be precise. (L44-45)

**6. L. 156, … Gaspari-Cohn function (GC; Eq. 4.10, Gaspari and Cohn, 1999) … It could be good to give the explicit expression of the function that you are referring to here.**
We think it is not necessary to add the explicit expression as the function is referenced. (L45)

**7. L. 116. Mention the year (2016, according to the caption of several Figures)**
We added the year. (L117)

**8. L. 131, … a sample of state vectors xn … (with explicit superscripts)**
We added the superscript. (L132)

**9. L. 186, … with unique members … , obscure. I understand without overlap.**
We rephrased the sentence to avoid confusion and relaced "unique" with "without repetition." (L187)

**10. L. 224, Eq. (5), mention that S is the number of 40-member ensembles (S = 25)**
We now include the definition of S. (L227)

**11. L. 271, The variability between day to night also appears to be small. That does not seem to be visible from what you show. Insert the words (not shown) or something similar. The same remark applies to other places, for instance to l. 357.**
We added "(not shown)" as suggested. (L270; L357)

**12. L. 366, what are the GC setups that are supposed to be listed in Sec. 2.3.1 ? Be more explicit, or remove.**
We slightly adapted section 3.3.3 to avoid confusion, and are now more explicit. (L365-371)

**13. L. 397, … derogates the error reduction. That makes no sense to me. Do you mean … increases the error. ?**
We replaced "derogates" with "leads to less" error reduction. (L397)

**14. Ll. 464-465, The shapes that we found included Gaussian, exponential or linear functions. Well, you have shown none of that.**
We agree that this comment was not precise and removed it. (L467-468)

**15. Fig. 7. The standard deviations on the All curves cannot be seen (whether on the screen or on a print-out). If standard deviations are so small that they cannot be distinguished from the (rather thick) curves themselves, say it (the same remark may apply to other figures, please check that aspect carefully).**
We checked all figures and added a comment to the caption of Fig 7 that refers to the small standard deviation. (see Fig7)

**16. L. 448, Our analysis includes three prognostic variables (humidity, temperature, and wind). Well, you have so far spoken of four variables, including the two wind components.**
We changed three to four and now mention the "horizontal wind components". (L450-451)

**Please revise your paper according to Lili Lei's comment, as well as to mine. In case you disagree with a particular comment, or decide not to follow a particular suggestion, please state precisely your reasons for that.**

**I thank you for having thought of Nonlinear Processes in Geophysics for submitting your paper, and look forward to receiving a further revised version.**
We addressed all comments and hope that the manuscript in its present form can be accepted for publication in NPG. Thank you!

**REFERENCE**
**Higham, N. J.: Computing the nearest correlation matrix—a problem from finance, IMA Journal of Numerical Analysis, 22, 329–343, https://doi.org/10.1093/imanum/22.3.329, 2002.**

---

## Author Response (AR4)

**Response to technical corrections - *egusphere-2022-434*:**

**Guidance on how to improve vertical covariance localization
based on a 1000-member ensemble**

**by Tobias Necker, David Hinger, Philipp Johannes Griewank, Takemasa Miyoshi, and Martin Weissmann**

We would like to thank the editor for accepting our manuscript for publication with technical corrections. Below we respond to how we addressed the technical corrections in the final manuscript. **Original queries are bold**, and our responses are normal text.

**EC4 Comments to the author:**
**The expression Gaussian-shaped is still to be removed in my opinion (ll. 157 and 373)**
We removed "Gaussian-shaped" as suggested.

**I suggest you modify your new comment in the caption of Fig.7 to "Note that the standard deviations are small and hardly distinguishable from the curves themselves."**
We modified the caption as suggested.